# Federated Time Series Generation on Feature and Temporally Misaligned Data

## Abstract

Distributed time series data presents a challenge for federated learning, as clients often possess different feature sets and have misaligned time steps. Existing federated time series models are limited by the assumption of perfect temporal or feature alignment across clients. In this paper, we propose FedTDD, a novel federated time series diffusion model that jointly learns a synthesizer across clients. At the core of FedTDD is a novel data distillation and aggregation framework that reconciles the differences between clients by imputing the misaligned timesteps and features. In contrast to traditional federated learning, FedTDD learns the correlation across clients' time series through the exchange of local synthetic outputs instead of model parameters. A coordinator iteratively improves a global distiller network by leveraging shared knowledge from clients through the exchange of synthetic data. As the distiller becomes more refined over time, it subsequently enhances the quality of the clients' local feature estimates, allowing each client to then improve its local imputations for missing data using the latest, more accurate distiller. Experimental results on five datasets demonstrate FedTDD's effectiveness compared to centralized training, and the effectiveness of sharing synthetic outputs to transfer knowledge of local time series. Notably, FedTDD achieves 79.4% and 62.8% improvement over local training in Context-FID and Correlational scores.

## 1 Introduction

Multivariate time series data are pivotal in many domains, such as healthcare, finance, manufacturing, and sales (Lim & Zohren, 2021). Consider a collaboration between multiple clients, shown in Figure 1. In a healthcare setting, these clients could be hospitals, each collecting patient data locally for a downstream task, such as predicting patient outcomes. The data gathered, such as vital signs like heart rate and blood pressure, is inherently *temporal*, i.e., time series data. Aggregating data from all the sources could improve model performance due to increased sampled diversity when training downstream predictive models. However, pri-

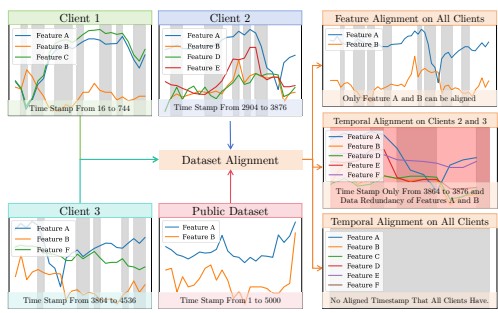

Figure 1: Feature and temporally misaligned time series. The gray masking indicates missing data.

vacy regulations such as the General Data Protection Regulation (GDPR) (Voigt & Von dem Bussche, 2017) and confidentiality agreements between hospitals prevent sharing of raw data (Alaa et al., 2021).

*Federated learning* (FL) (McMahan et al., 2017) takes a step towards tackling this privacy challenge by enabling clients to train a global model by sharing locally trained model parameters rather than raw data. However, this environment faces the challenge of *feature and temporal misalignment* (Luu et al., 2021), as hospitals may possess different feature sets with varying time intervals for data collection.

In *horizontal* FL (Li et al., 2020), different clients have data for the same features but for different samples or timesteps. Hence, it can tackle situations involving temporal misalignment but not feature misalignment. On the other hand, in *vertical* FL (Liu et al., 2024), different clients possess different feature sets for the same samples or timesteps. While this can handle feature misalignment, it cannot tackle temporal misalignment. Hence, neither horizontal nor vertical FL can fully tackle scenarios with both feature and temporal misalignment. On top of this, data may be missing or incomplete due to unavailability or inconsistent collection frequencies, further hindering a model's ability to learn patterns (Pratama et al., 2016).

To overcome these limitations, we propose **FedTDD** (Federated Learning in Multivariate Time Series via Data Distillation), a first-of-its-kind federated time series diffusion model capable of learning a time series synthesizer from clients' distinct features with temporal misalignment. FedTDD introduces a novel data distillation (Sachdeva & McAuley, 2023) and aggregation framework for the common feature set, whose values differ across clients and can be obtained from the public domain. In this framework, a coordinator maintains a global model called the *distiller*, trained iteratively using a combination of public data and clients' intermediate synthetic data outputs. Each client keeps a local time series diffusion model for imputing local features which leverages the latest distiller to improve the quality of local estimates. Unlike traditional federated learning, FedTDD learns the correlations among clients' time series through the exchange of synthetic outputs instead of aggregating models (McMahan et al., 2017), effectively handling feature and temporal misalignment without sharing raw data.

Given the recent advancements of diffusion models over mainstream generative models like *Generative Adversarial Networks* (GANs) (Goodfellow et al., 2020), we utilize a time series *Denoising Diffusion Probabilistic Model* (DDPM) (Ho et al., 2020), adapted to handle temporal dependencies through temporal embeddings and sequential conditioning. Specifically, we select **Diffusion-TS** (Yuan & Qiao, 2024) since it leverages both time and frequency domain information, effectively capturing trends and seasonality, which leads to a more accurate imputation of missing data. By imputing data from unaligned time steps, clients can obtain temporally aligned data without needing alignment on the features or sharing raw data.

In summary, our major contributions are as follows: (i) We propose a novel federated generative learning framework that effectively handles temporal and feature-level misalignment and data missing problems in time series data. (ii) We develop a data distillation and aggregation framework that learns correlations among clients' time series by exchanging synthetic data instead of model parameters, enabling clients to improve their local models without direct data sharing and effectively handling data discrepancies. (iii) We conduct extensive experiments on five benchmark datasets, showing up to 79.4% and 62.8% improvement over local training in Context-FID and Correlational scores under extreme feature and temporal misalignment cases and achieving performance comparable to centralized training.

## 2 RELATED WORK

Table 1: Overview of the related work.

| Method | Model Type | Time Series | FL Type | Handles Temporal Misalignment | Handles Feature Misalignment |
|---|---|---|---|---|---|
| GTV (Zhao et al., 2023) | GAN | × | Vertical | × | ✓ |
| DPGDAN (Wang et al., 2023) | GAN | × | Vertical | × | ✓ |
| SiloFuse (Shankar et al., 2024) | DDPM | × | Vertical | × | ✓ |
| VFLGAN-TS (Yuan et al., 2024) | GAN | ✓ | Vertical | × | ✓ |
| FedGAN (Rasouli et al., 2020) | GAN | ✓ | Horizontal | ✓ | × |
| T2TGAN (Brophy et al., 2021) | GAN | ✓ | Horizontal | ✓ | × |
| FedTDD (Ours) | DDPM | ✓ | Hybrid | ✓ | ✓ |

**Time series generation** Generative models for time series data aim to capture temporal dependencies and sequential patterns inherent in such datasets. TimeGAN (Yoon et al., 2019) combines *generative adversarial networks* (GANs) Goodfellow et al. (2020) with recurrent neural networks (Mogren, 2016) to produce realistic multivariate time series. TimeVAE (Desai et al., 2021) utilizes variational autoencoders (VAEs) (Kingma, 2013) tailored for time series to capture trends and sea-

sonality. Recently, diffusion-based models like TimeGrad (Rasul et al., 2021), CSDI (Tashiro et al., 2021), SSSD (Alcaraz & Strodthoff, 2022), TSDiff (Kollovieh et al., 2024), and Diffusion-TS (Yuan & Qiao, 2024) have further advanced time series generation by producing high-fidelity sequences, outperforming the mainstream GANs and VAE-based techniques. Despite their effectiveness, these models operate in centralized settings and assume fully aligned data with consistent features and timestamps. They are not equipped to handle feature and temporal misalignments common in real-world distributed scenarios, making them unsuitable for federated environments with heterogeneous data distributions (Mendieta et al., 2022; Qu et al., 2022; Ye et al., 2023).

**Federated learning with generative models**    Federated learning (Zhang et al., 2021) has primarily been applied to image generation, such as FedCycleGAN (Song & Ye, 2021) leverages Cycle-GAN (Zhu et al., 2017) in federated settings to generate synthetic images while preserving data privacy. For tabular data, methods like GTV (Zhao et al., 2023), DPGDAN (Wang et al., 2023), and SiloFuse (Shankar et al., 2024) employ GANs and diffusion models within vertical federated learning frameworks to synthesize tabular datasets. However, these approaches focus on vertically partitioned data, where all clients have features corresponding to the same sample ID, and do not address data redundancy or misalignment issues. Federated learning with generative models for time series data remains under-explored. Existing works such as FedGAN (Rasouli et al., 2020), VFLGAN-TS (Yuan et al., 2024), and T2TGAN (Brophy et al., 2021) extend GANs to federated time series generation. VFLGAN-TS operates in a vertical federated learning context, tackling feature misalignment, but does not handle temporal misalignment. In contrast, T2TGAN tackles horizontal federated learning settings but introduces data redundancy due to overlapping data among clients and cannot handle feature mismatches between clients. As summarized in Table 1, these methods encounter issues as shown in Figure 1, making them less effective for federated time series generation where both feature and temporal misalignments are prevalent.

**Preliminary on generative modeling with DDPMs**    For the generative backbone, we adopt the Diffusion-TS architecture (Yuan & Qiao, 2024), which extends DDPMs Ho et al. (2020) to capture temporal patterns using a generative modeling process. DDPMs are models trained using a *forward noising* and *backward denoising* process. The forward phase progressively adds random Gaussian noise to the data $\mathbf{s}_0$ at diffusion step $t$, where the transition is parameterized by $q(\mathbf{s}_t \mid \mathbf{s}_{t-1}) = \mathcal{N}(\mathbf{s}_t; \sqrt{1 - \beta_t}\,\mathbf{s}_{t-1}, \beta_t \mathbf{I})$ with $\beta_t \in (0, 1)$, eventually transforming it into pure noise $\mathbf{s}_T \sim \mathcal{N}(0, \mathbf{I})$. The backward phase is where the model learns to reverse this noising process. Starting from random noise $\mathbf{s}_T \sim \mathcal{N}(0, \mathbf{I})$, it iteratively removes the added noise step by step via $p_\theta(\mathbf{s}_{t-1} \mid \mathbf{s}_t) = \mathcal{N}(\mathbf{s}_{t-1}; \boldsymbol{\mu}_\theta(\mathbf{s}_t, t), \Sigma_\theta(\mathbf{s}_t, t))$, to reconstruct a new data sample resembling the original input distribution. The functions $\boldsymbol{\mu}_\theta$ and $\Sigma_\theta$ are generally estimated using a model.

Diffusion-TS extends standard DDPMs by incorporating mechanisms specifically designed for time series characteristics such as trends and seasonality (Kitagawa & Gersch, 1984). Instead of treating data points independently, it utilizes an encoder-decoder transformer architecture (Vaswani, 2017) that processes entire sequences, effectively modeling temporal relationships. To handle trends, Diffusion-TS decomposes the time series into components that represent slow-varying behaviors over time. For capturing seasonality and periodic patterns, it employs frequency domain analysis using the Fast Fourier Transform (FFT) (Cooley et al., 1969; Heckbert, 1995). By integrating FFT, the model can analyze and reconstruct cyclical patterns (Ceneda et al., 2018) within the data, allowing it to learn both time and frequency domain representations (Fons et al., 2022). This combination enables Diffusion-TS to generate more accurate and realistic time series data by effectively modeling complex temporal dynamics. Besides, Diffusion-TS supports both *unconditional* and *conditional* generation. In the unconditional generation, the model produces new samples solely based on the learned data distribution, starting from random noise and applying the learned denoising process. In the conditional generation, Diffusion-TS utilizes gradient-based guidance during sampling to incorporate the observed data $\mathbf{y}$. At each diffusion step, the model refines its estimated time series $\hat{\mathbf{s}}_0$ by adjusting it with a gradient term that enforces consistency with the observed data. The refinement can be computed via $\tilde{\mathbf{s}}_0(\mathbf{s}_t, t; \theta) = \hat{\mathbf{s}}_0(\mathbf{s}_t, t; \theta) + \eta \nabla_{\mathbf{s}_t}(\|\mathbf{y} - \hat{\mathbf{s}}_0(\mathbf{s}_t, t; \theta)\|^2 + \gamma \log p(\mathbf{s}_{t-1} \mid \mathbf{s}_t))$, where $\eta$ is a hyperparameter that controls the strength of the gradient guidance, and $\gamma$ balances the trade-off between fitting the observed data and maintaining the generative model's prior distribution $p(\mathbf{s}_{t-1} \mid \mathbf{s}_t)$. This iterative refinement ensures that the generated time series aligns with the provided observations and preserves the temporal patterns learned during training. Further details of Diffusion-TS are shown in Appendix B.2.

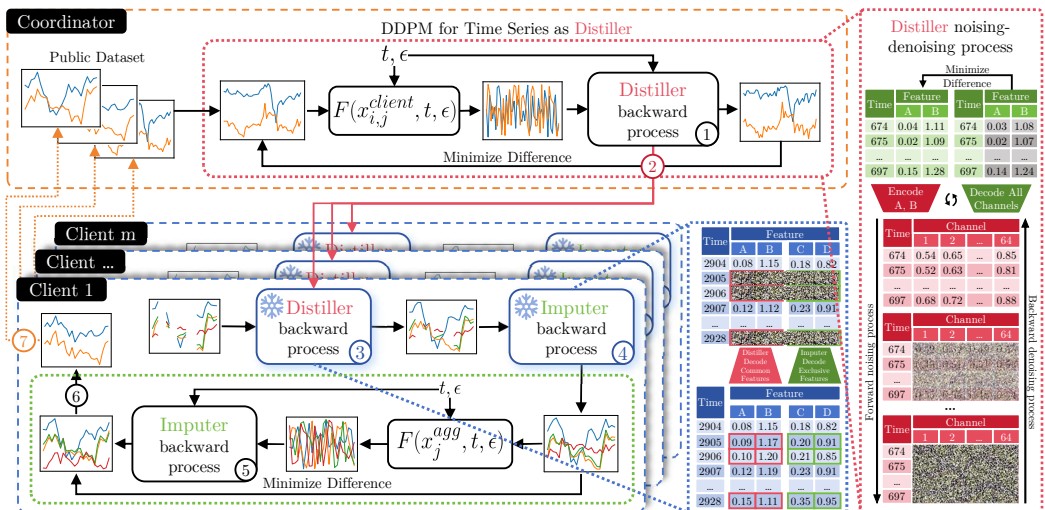

Figure 2: FedTDD Structure. First, the Distiller is pre-trained on a public dataset. Then, each client uses the distiller and imputer to impute common and exclusive features, respectively. Finally, synthetic data is sent back to the coordinator to expand the public dataset for the next round. The order of execution(1-7) is labeled in the figure. Here the common features are A and B, and the exclusive features are C and D.

## 3 FedTDD

In this work, we address the problem of collaborative time series imputation in the presence of temporal and feature misalignments, without requiring the sharing of raw data. In a federated learning setting, clients may possess different subsets of features. We categorize features into two types: *common features* and *exclusive features*. Common features are those present in all clients and also available in a public dataset, while exclusive features are unique to each client and not shared. For example, market indices might be common features in financial data, while individual portfolio holdings are exclusive. Our proposed framework, FedTDD, as shown in Figure 2, tackles this problem using two models. A global distiller first imputes missing common features across clients. Local imputer models then use the imputed common features to predict the missing exclusive features for each client, addressing both temporal and feature misalignments. Furthermore, clients protect their privacy by sharing only synthetic versions of the common features while collaboratively improving the global distiller. This cycle of iterative imputation and model refinement ultimately converges to yield good quality imputations, while ensuring that no raw data is shared.

### 3.1 PROBLEM DEFINITION

We consider a federated learning setup involving $N$ clients and a coordinator. Each client $i$ possesses a time series dataset, denoted as $\mathbf{X}^i = \left[ X^i_{j,k} \right]_{\{j=1...T^i, k=1...C^i\}}$, where $T^i$ is the number of time steps, and $C^i$ is the number of channels. These datasets can be split into two components, one for the common features and one for the exclusive features, i.e., $\mathbf{X}^i = \mathbf{X}^i_{\text{comm}} \cup \mathbf{X}^i_{\text{ex}}$. The coordinator holds a public dataset $\mathbf{X}^{\text{pub}} = \left[ X^{\text{pub}}_{j,k} \right]_{\forall j; k \in \mathcal{F}_{\text{comm}}}$, which contains data for the common features $\mathcal{F}_{\text{comm}}$ but without any missing values. This public dataset is time-indexed differently from the clients' data and provides a reliable reference for the common features. Each client's time series data comes from a distinct time interval, meaning that each client's time indices $j$ are unique. The feature set for each client $i$, $\mathcal{F}^i$, consists of common features $\mathcal{F}_{\text{comm}}$, which are shared across all clients, and exclusive features $\mathcal{F}^i_{\text{ex}}$, which are specific to each client. Thus, the overall feature set for client $i$ is represented as $\mathcal{F}^i = \mathcal{F}_{\text{comm}} \cup \mathcal{F}^i_{\text{ex}}$. Conversely, clients may have missing values in both the common and exclusive features. These missing values are indicated by a binary mask matrix $\mathbf{M}^i = \left[ M^i_{j,k} \right]_{\forall j,k}$,

---

**Algorithm 1:** FedTDD

---

**Input:** Public dataset $\mathbf{X}^{\text{pub}}$, clients' datasets $\mathbf{X}^i$
**Result:** Global distiller model $\mathcal{D}$, local imputer models $\mathcal{U}^i$

**1 Initialize:** Train $\mathcal{D}$ on $\mathbf{X}^{\text{pub}}$
**2 for** $r = 1$ *to* $R$ **do**
**3**    **for** *each client $i$* **do**
**4**       **Receive** global distiller $\mathcal{D}$
**5**       $\hat{\mathbf{X}}^i_{\text{comm}} \leftarrow \mathcal{D}\left(\mathbf{X}^i_{\text{comm}}, \mathbf{M}^i_{\text{comm}}\right)$ ;                    ▷ Impute common features
**6**       $\hat{\mathbf{X}}^i_{\text{ex}} \leftarrow \mathcal{U}\left(\mathbf{X}^i_{\text{ex}}, \mathbf{M}^i_{\text{ex}}\right)$ ;                       ▷ Impute exclusive features
**7**       $\mathbf{X}^i_{\text{train}} \leftarrow \hat{\mathbf{X}}^i_{\text{comm}} \cup \hat{\mathbf{X}}^i_{\text{ex}}$ ;             ▷ Combine with exclusive features
**8**       **Train** $\mathcal{U}^i$ on $\mathbf{X}^i_{\text{train}}$ ;                       ▷ Train local imputer
**9**       $\hat{\mathbf{X}}^i \leftarrow \mathcal{U}^i(\mathbf{z}), \mathbf{z} \sim \mathcal{N}(0, \mathbf{I})$ ;           ▷ Generate synthetic data
**10**       **Send** $\hat{\mathbf{X}}^i_{\text{comm}}$ from $\hat{\mathbf{X}}^i$ to coordinator
**11**    **end**
**12**    **for** *each client $i$* **do**
**13**       Select $n_r = \dfrac{r}{R} \alpha \cdot L$ sequences from $\hat{\mathbf{X}}^i_{\text{comm}}$
**14**       $\mathbf{X}^{\text{pub}} \leftarrow \mathbf{X}^{\text{pub}} \cup \hat{\mathbf{X}}^i_{\text{comm}}[1 : n_r]$ ;         ▷ Expand public dataset
**15**    **end**
**16**    **Finetune** $\mathcal{D}$ on updated $\mathbf{X}^{\text{pub}}$
**17 end**

---

where $M^i_{j,k} = 1$ if the value $X^i_{j,k}$ is observed while $0$ indicates it is missing. The mask can be split into two parts: $\mathbf{M}^i_{\text{comm}}$, which corresponds to missing data in the common features, and $\mathbf{M}^i_{\text{ex}}$, which corresponds to missing data in the exclusive features. The goal is to design a collaborative method that enables clients to leverage shared knowledge and the public dataset to input the missing data locally without sharing raw data. Table 4 summarises the mathematical notations used.

### 3.2 Hybrid Federated Learning for imputation under misalignment

Algorithm 1 presents the overview of FedTDD. The framework consists of two key components: the global distiller model $\mathcal{D}$ and the local imputer models $\mathcal{U}^i$. The global distiller $\mathcal{D}$ imputes missing common features shared across all clients, while each client trains a local imputer $\mathcal{U}^i$ to infer missing exclusive features specific to their data. These components work together to address temporal and feature misalignment by iteratively improving the imputation process over several rounds $r$ ranging from 1 to $R$.

The process begins with the coordinator training a global distiller model $\mathcal{D}$ using the public dataset $\mathbf{X}^{\text{pub}}$. $\mathcal{D}$ leverages a temporal DDPM backbone to apply a forward diffusion process by gradually adding noise to the data and learns to reverse this process. During training, $\mathcal{D}$ conducts *unconditional generation* by starting from Gaussian noise $\boldsymbol{\epsilon}$ and learning to approximate the data distribution through the time and frequency domain components (Yuan & Qiao, 2024). Formally, we have

$$\mathcal{L}_{\text{time}} = \mathbb{E}_{(j,k,t) \mid \mathbf{M}^{\text{pub}}_{j,k}=1} \left[ \left\| \mathbf{X}^{\text{pub}}_{j,k} - \tilde{\mathbf{X}}^{\text{pub}}_{j,k}(\mathbf{X}^{\text{pub}}_{j,k,t}, t, \boldsymbol{\epsilon}; \theta) \right\|^2 \right] \quad \text{and} \tag{1}$$

$$\mathcal{L}_{\text{freq}} = \mathbb{E}_{(j,k,t) \mid \mathbf{M}^{\text{pub}}_{j,k}=1} \left[ \left\| \text{FFT}(\mathbf{X}^{\text{pub}}_{j,k}) - \text{FFT}\left( \tilde{\mathbf{X}}^{\text{pub}}_{j,k}(\mathbf{X}^{\text{pub}}_{j,k,t}, t, \boldsymbol{\epsilon}; \theta) \right) \right\|^2 \right], \tag{2}$$

where $\boldsymbol{\epsilon} \sim \mathcal{N}(0, \mathbf{I})$, $\mathbf{X}^{\text{pub}}_{j,k}$ is the $(j, k)$-th entry of $\mathbf{X}^{\text{pub}}$, $\tilde{\mathbf{X}}^{\text{pub}}_{j,k}$ is the denoised estimate from $\mathcal{D}$, and FFT denotes the Fast Fourier Transform (Heckbert, 1995), which is a mathematical operation that converts a finite-length time domain signal to its frequency domain representation. We take the following objective

$$\mathcal{L}_{\text{distiller}(\mathcal{D}^i)} = \mathbb{E}_{(j,k,t) \mid \mathbf{M}^{\text{pub}}_{j,k}=1} \left[ w_t \left( \lambda_1 \mathcal{L}_{\text{time}} + \lambda_2 \mathcal{L}_{\text{freq}} \right) \right], \quad w_t = \frac{\lambda \gamma_t (1 - \bar{\gamma}_t)}{\delta_t^2}, \tag{3}$$

where $\lambda_1$ and $\lambda_2$ control the balance between time and frequency losses while $w_t$ emphasizes learning at larger diffusion steps, with $\lambda$ being a small constant. The parameter $\delta_t \in (0, 1)$ determines

the amount of noise added at each forward diffusion step, where $t$ is a diffusion time step uniformly sampled from 1 to $T$ during training. The cumulative product $\bar{\gamma}_t = \prod_{v=1}^{t} \gamma_v$, with $\gamma_t = 1 - \delta_t$, track how the original signal diminishes over time due to the added noise. By weighting the loss at different steps, $w_t$ helps the model focus on reconstructing the signal under high-noise conditions.

After this initial training, the coordinator distributes the trained global distiller model $\mathcal{D}$ to all participating clients. Each client $i$ then utilizes $\mathcal{D}$ to impute their missing common features. Since clients may have missing values in $\mathbf{X}_{\text{comm}}^{i}$, they input their data along with the corresponding mask $\mathbf{M}_{\text{comm}}^{i}$ to the distiller model, which will perform *conditional generation* to iteratively refine the imputed data by sampling from the conditional distribution guided by the observed data, shown in Equation 23. The imputation process follows $\hat{\mathbf{X}}_{\text{comm}}^{i} = \mathcal{D}(\mathbf{X}_{\text{comm}}^{i}, \mathbf{M}_{\text{comm}}^{i})$, where $\mathcal{D}$ reconstructs only the missing values, indicated by $\mathbf{M}_{\text{comm}}^{i} = 0$. Similarly, the local imputer imputes missing values in $\mathbf{X}_{\text{ex}}^{i}$ by inputting their data along with the corresponding mask $\mathbf{M}_{\text{ex}}^{i}$ to the imputer model via $\hat{\mathbf{X}}_{\text{ex}}^{i} = \mathcal{U}(\mathbf{X}_{\text{ex}}^{i}, \mathbf{M}_{\text{ex}}^{i})$. The imputed common features $\hat{\mathbf{X}}_{\text{comm}}^{i}$ are then combined with the available exclusive features $\hat{\mathbf{X}}_{\text{ex}}^{i}$ to form the training data $\mathbf{X}_{\text{train}}^{i} = \hat{\mathbf{X}}_{\text{comm}}^{i} \cup \hat{\mathbf{X}}_{\text{ex}}^{i}$ for the local imputer. Meanwhile, each client trains their local imputer model $\mathcal{U}^i$ using $\mathbf{X}_{\text{train}}^{i}$ as the ground truth. Since the imputed common features $\hat{\mathbf{X}}_{\text{comm}}^{i}$ are fully known (as they are outputs from the pre-trained and fine-tuned $\mathcal{D}$), they are entirely used as ground truth for training $\mathcal{U}^i$, regardless of the original mask $\mathbf{M}_{\text{comm}}^{i}$. For the exclusive features, only the observed entries indicated by the mask $\mathbf{M}_{\text{ex}}^{i}$ are used as ground truth since the quality of the imputer's generated data during training is not sufficient to be used as ground truth. We define the loss mask as $\mathbf{M}_{\text{loss}}^{i} = \mathbf{1}_{\text{comm}}^{i} \cup \mathbf{M}_{\text{ex}}^{i}$, where $\mathbf{1}_{\text{comm}}^{i}$ is a matrix of ones corresponding to the common features of client $i$. This loss mask ensures that the reconstruction loss is computed over all entries of the imputed common features and the observed entries of the exclusive features. The training loss for the imputer $\mathcal{U}^i$ can be defined as follows:

$$\mathcal{L}_{\text{imputer}}(\mathcal{U}^i) = \mathbb{E}_{(j,k,t) \,|\, \mathbf{M}_{\text{loss}_{j,k}}^{i}=1} \left[ w_t \left( \lambda_1 \mathcal{L}_{\text{time}}^{i} + \lambda_2 \mathcal{L}_{\text{freq}}^{i} \right) \right], \tag{4}$$

$$\text{where} \quad \mathcal{L}_{\text{time}}^{i} = \mathbb{E}_{(j,k,t) \,|\, \mathbf{M}_{\text{loss}_{j,k}}^{i}=1} \left[ \left\| \mathbf{X}_{\text{train}_{j,k}}^{i} - \tilde{\mathbf{X}}_{\text{train}_{j,k}}^{i}(\mathbf{X}_{\text{train}_{j,k,t}}^{i}, t; \theta) \right\|^2 \right] \tag{5}$$

$$\text{and} \quad \mathcal{L}_{\text{freq}}^{i} = \mathbb{E}_{(j,k,t) \,|\, \mathbf{M}_{\text{loss}_{j,k}}^{i}=1} \left[ \left\| \text{FFT}(\mathbf{X}_{\text{train}_{j,k}}^{i}) - \text{FFT}\left( \tilde{\mathbf{X}}_{\text{train}_{j,k}}^{i}(\mathbf{X}_{\text{train}_{j,k,t}}^{i}, t; \theta) \right) \right\|^2 \right], \tag{6}$$

where $\mathbf{X}_{\text{train}_{j,k}}^{i}$ is the $(j,k)$-th entry of $\mathbf{X}_{\text{train}}^{i}$, $\tilde{\mathbf{X}}_{\text{train}_{j,k}}^{i}$ is the denoised estimate from $\mathcal{U}$. After training, each client uses the trained imputer $\mathcal{U}^i$ to generate a synthetic dataset through *unconditional synthesis*, which includes both the common features $\hat{\mathbf{X}}_{\text{comm}}^{i}$ and the exclusive features $\hat{\mathbf{X}}_{\text{ex}}^{i}$. Starting from Gaussian noise, the imputer generates samples $\hat{\mathbf{X}}^{i} = \mathcal{U}^i(\mathbf{z}), \mathbf{z} \sim \mathcal{N}(0, \mathbf{I})$, that capture the distribution of both common and exclusive features.

To protect privacy, only the common features from the synthetic dataset, $\hat{\mathbf{X}}_{\text{comm}}^{i}$, are shared with the coordinator. This ensures that no raw or exclusive client data is exposed during the collaborative learning. The coordinator uses the synthetic common feature data from the clients to expand its public dataset. Rather than simply absorbing all the synthetic data, the coordinator carefully controls the growth of the dataset by accepting a fraction of the sequences from each client. Specifically, the coordinator adds $\frac{r}{R}\alpha * L$, where $L$ represents the length of the synthetic datasets $\hat{\mathbf{X}}_{\text{comm}}^{i}; \forall i \in \{1, 2, \ldots, N\}$, $\alpha$ is a hyperparameter between 0 and 1, and the ratio of $r$ and $R$ yields a number that linearly increases up to 1, allowing for a gradual expansion as the rounds increase. The coordinator retrains the global distiller $\mathcal{D}$ using this expanded dataset. The addition of synthetic data enhances the distiller's ability to learn the patterns necessary for imputing missing common features.

The overall process creates an iterative cycle of improvement. As clients' generative models, specifically their local imputers, become more accurate with each round, the quality of the synthetic data they generate also improves. This higher-quality synthetic data, in turn, improves the distiller model at the coordinator, which benefits all clients when it is redistributed. Over several training rounds, this mutual reinforcement drives both the global distiller and the local imputers to improve continuously. Ultimately, the process converges, yielding robust imputation models without requiring clients to share their raw data.

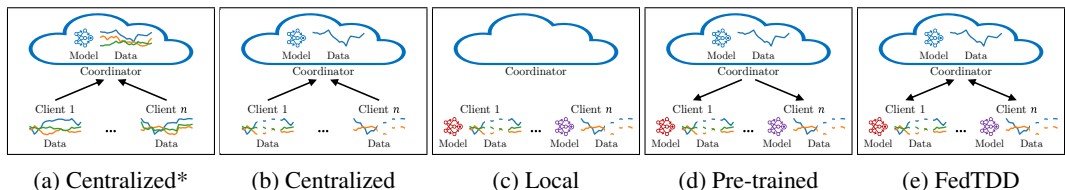

| (a) Centralized* | (b) Centralized | (c) Local | (d) Pre-trained | (e) FedTDD |

Figure 3: Illustrations of different baselines compared to FedTDD. The data in the coordinator, also called public data, in Figure 3b, 3d and 3e consists only common features time series. Dashes indicate temporal missing values.

## 4 EXPERIMENTS

We assess FedTDD's performance by showing its advantages and disadvantages when applied to multiple benchmark datasets. We leave the analysis of different training configurations in the Appendix C, where we examine the impact of limited public data, abundant sequences with missing data, imbalanced data distributions and different aggregation strategies on model performance.

**Datasets**  To assess the quality of synthetic data, we consider four real-world datasets and one simulated dataset with different properties, such as the number of features, correlation, periodicity, and noise levels. Each dataset is preprocessed using a sliding window technique (Yoon et al., 2019) to segment the data into sequences of length 24 to capture meaningful temporal dependencies while keeping the computational cost manageable. **Stocks** (Yoon et al., 2019) is the daily historical Google stock data from 2004 to 2019 with highly correlated features. **ETTh** (Zhou et al., 2021) recorded the electricity transformers hourly between July 2016 and July 2018, including load and oil temperature data that consists of 7 features. **Energy** (Candanedo, 2017) from UCI appliances energy prediction dataset with 10-minute intervals for about 4.5 months. **fMRI** (Smith et al., 2011) is a realistic simulation of brain activity time series with 50 features. **MuJoCo** (Tunyasuvunakool et al., 2020) is a physics-based simulation time series containing 14 features. We show the statistics of all datasets in Appendix D.2.

**Baselines**  We compare FedTDD against approaches show in Figure 3a, 3b, 3c and 3d. For the **Centralized\*** training, we aggregate all data from individual clients, including public data, into a single location, where a global model is trained using the combined dataset, and this will be trained with all available features in the dataset and without missing values. While **Centralized** uses the same training procedure as Centralized*, it is, however, trained on a combined dataset with missing values and corresponding features available from each client plus the public data. To deal with differing features across clients, we create the combined dataset consisting of the total number of features in the particular benchmark dataset and zero-fill any remaining features to ensure uniformity. On the other hand, **Local** training involves training a separate model for each client using only their local data, without any communication or data aggregation. This approach has to be done to verify that FedTDD can perform relatively better than train locally. Finally, the **Pre-trained** approach leverages a model trained on a public dataset and uses it to impute the common features in local data from each client. Again, there is no data aggregation for this approach. In comparison, FedTDD integrates the Pre-trained approach and applies data aggregation to it. We utilized a SOTA diffusion-based multivariate time series generative model, Diffusion-TS (Yuan & Qiao, 2024), as the backbone for these baselines and FedTDD. Alternatively, any other time series generative model can be adopted in these approaches in a plug-and-play manner.

**Evaluation metrics**  We quantitatively assess the quality of the generated synthetic data using four key metrics (see Appendix D.3 for more details). **Context-Fréchet Inception Distance (Context-FID) score** (Jeha et al., 2022) evaluates the similarity between the distribution of real and synthetic time series data by computing the Fréchet distance. **Correlational score** (Liao et al., 2020) measures the correlation between the features of multivariate time series in the synthetic data compared to its real data. **Discriminative score** (Yoon et al., 2019) measures the realism of the synthetic data by training a binary classifier to distinguish between real and synthetic data. **Predictive score** (Yoon et al., 2019) evaluates the utility of the synthetic data by training a sequence-to-sequence model on

Table 2: Results on multiple time series datasets. **Bold** indicates best performance.

| Metric | Method | Stocks | ETTh | MuJoCo | Energy | fMRI |
|---|---|---|---|---|---|---|
| Context-FID | Centralized* | 0.682 +/- 0.106 | 0.281 +/- 0.040 | 0.782 +/- 0.138 | 0.533 +/- 0.082 | 1.737 +/- 0.125 |
| | Centralized | 3.548 +/- 0.990 | 8.870 +/- 2.295 | 10.00 +/- 2.814 | 9.343 +/- 2.808 | 13.56 +/- 3.357 |
| | Local | 1.648 +/- 0.229 | 1.313 +/- 0.188 | 0.751 +/- 0.121 | 1.179 +/- 0.179 | 1.694 +/- 0.153 |
| | Pre-trained | 1.047 +/- 0.169 | 0.326 +/- 0.040 | 0.617 +/- 0.090 | 0.412 +/- 0.054 | **1.411 +/- 0.102** |
| | FedTDD | **0.675 +/- 0.087** | **0.271 +/- 0.038** | **0.529 +/- 0.068** | **0.376 +/- 0.056** | 1.459 +/- 0.099 |
| Correlational | Centralized* | 0.061 +/- 0.043 | 0.253 +/- 0.094 | 1.989 +/- 0.247 | 5.231 +/- 1.294 | 7.900 +/- 0.384 |
| | Centralized | 0.769 +/- 0.336 | 0.340 +/- 0.097 | 2.230 +/- 0.518 | 5.681 +/- 0.634 | 18.07 +/- 2.311 |
| | Local | 0.156 +/- 0.120 | 0.239 +/- 0.079 | 1.298 +/- 0.260 | 3.447 +/- 0.838 | **5.992 +/- 0.383** |
| | Pre-trained | 0.077 +/- 0.052 | 0.165 +/- 0.074 | 1.323 +/- 0.171 | 2.821 +/- 0.651 | 6.049 +/- 0.349 |
| | FedTDD | **0.058 +/- 0.050** | **0.161 +/- 0.064** | **1.296 +/- 0.215** | **2.800 +/- 0.686** | 6.017 +/- 0.364 |
| Discriminative | Centralized* | 0.136 +/- 0.091 | 0.199 +/- 0.061 | 0.297 +/- 0.108 | 0.230 +/- 0.080 | 0.422 +/- 0.074 |
| | Centralized | 0.476 +/- 0.042 | 0.475 +/- 0.017 | 0.474 +/- 0.024 | 0.496 +/- 0.006 | 0.477 +/- 0.030 |
| | Local | 0.340 +/- 0.153 | 0.298 +/- 0.060 | 0.200 +/- 0.092 | 0.329 +/- 0.087 | **0.397 +/- 0.061** |
| | Pre-trained | **0.175 +/- 0.117** | 0.115 +/- 0.060 | 0.208 +/- 0.068 | **0.141 +/- 0.068** | 0.419 +/- 0.051 |
| | FedTDD | 0.185 +/- 0.105 | **0.106 +/- 0.061** | **0.153 +/- 0.120** | 0.153 +/- 0.072 | 0.414 +/- 0.051 |
| Predictive | Centralized* | 0.040 +/- 0.000 | 0.127 +/- 0.003 | 0.112 +/- 0.015 | 0.292 +/- 0.009 | 0.137 +/- 0.004 |
| | Centralized | 0.047 +/- 0.012 | 0.223 +/- 0.020 | 0.165 +/- 0.060 | 0.427 +/- 0.053 | 0.233 +/- 0.051 |
| | Local | 0.043 +/- 0.003 | 0.118 +/- 0.011 | 0.048 +/- 0.006 | 0.204 +/- 0.012 | 0.135 +/- 0.006 |
| | Pre-trained | 0.046 +/- 0.001 | 0.104 +/- 0.004 | 0.052 +/- 0.004 | 0.177 +/- 0.005 | 0.133 +/- 0.006 |
| | FedTDD | **0.041 +/- 0.001** | **0.101 +/- 0.004** | **0.048 +/- 0.004** | **0.175 +/- 0.006** | **0.133 +/- 0.004** |

the synthetic data and measuring its performance on real data. All evaluation metrics are computed based on the respective features of the individual clients and then averaged over five trials, followed by calculating the overall average across the number of clients. The quality of synthetic data is considered the "best" when all metrics approach 0, meaning lower values indicate better quality.

**Training configurations** We run FedTDD and the baselines mentioned above with ten clients, five global rounds, 7500 local epochs for the first round, and 5000 for the rest. Besides, the coordinator trains on the public data consisting of common features, and each client contributes a set of features, which is the combination of common and exclusive features. The number of common features is around 50% of the total number of features in the original dataset. On the other hand, we use public ratio (**PR**) to manipulate the proportion of the public data that has to be reserved from the entire dataset before partitioning the dataset to all clients. Split ratio (**SR**) divides all sequences into two groups. In the first group, a mask is applied to just the common features, while in the second group, the mask is applied to all features. Moreover, missing ratio (**MR**) is the missing rate to mask on a sequence of multivariate time series, and we consider the missing scenario as shown in Appendix D.4. In the main experiments, we set **PR**, **SR**, and **MR** to 0.5. All the hyperparameters are listed in Appendix D.5.

## 4.1 TIME SERIES GENERATION

In Table 2, we quantitatively analyze the quality of unconditionally generated 24-length time series for diverse time series datasets. FedTDD shows a strong performance comparable to the Centralized* approach. The proposed aggregation mechanism during fine-tuning proved essential to prevent the degradation of the coordinator model's performance and, in turn, the client models. By doing this, we achieved strong results across most datasets. We also present the generated synthetic samples of one representative client for ETTh and fMRI datasets in Figure 4.

**Challenges on fMRI dataset** We observe that the fMRI dataset's imputation quality was lower than other datasets, as the mean square error between the imputed and real data is greater. Consequently, client models degraded due to training on low-quality imputed data. This suggests that the imputation strategy may need further refinement for such datasets, where the data distribution and complexity present greater challenges for accurate synthetic data generation and imputation.

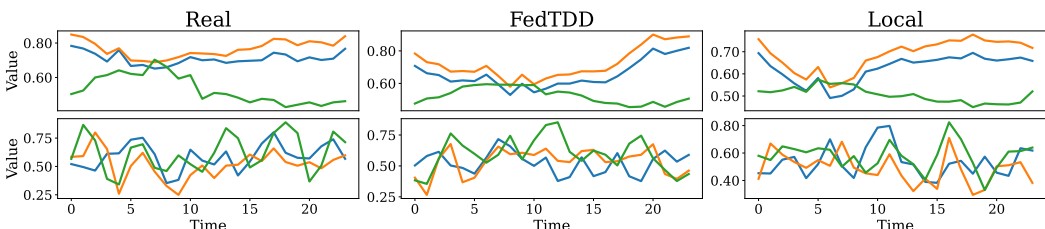

Figure 4: Real samples and synthetic samples generated unconditionally from FedTDD and Local. The first and second rows of samples are from ETTh and fMRI datasets, respectively.

Besides, the Local approach achieves the best Correlational and Discriminative scores for the fMRI dataset. However, we cannot conclude that training locally is the best overall approach for fMRI. As we mentioned, the low performance of FedTDD and Pre-trained is primarily due to the poor quality of the imputed data, which affects training. This shows the advantage of Local training not relying on imputed data, making it seem better suited for the fMRI dataset compared to FedTDD and Pre-trained.

**Comparison between Centralized and Local training**  Both Centralized and Local approaches are trained on datasets with missing values, but their performance differs significantly. This could be due to the different model architectures used in each approach. As aforementioned, the Centralized model is trained on a combined dataset where the additional features are filled with zeros, which results in the worst performance. This shows the advantage of having an individual model trained locally for each client.

### 4.2 ABLATION STUDY

In Table 3, we show the result of reducing the number of common features in FedTDD. We set the number of common features to around 25% of the total number of features in the corresponding dataset. As a result, we can observe the robustness of FedTDD when dealing with a relatively small number of common features across most datasets. However, FedTDD does not perform as expected on the fMRI dataset because of the poor quality of imputed data, as mentioned in Section 4.1. On the other hand, the performance of Centralized training slightly decreased due to more zeros filling out the combined dataset, especially in the public data.

### 5 CONCLUSION

While federated learning is increasingly applied for different regreasing tasks for time series (TS), it is still limited in handling generative tasks, especially when time series features are vertically partitioned and temporarily misaligned. We propose a novel federated TS generation framework, FedTDD, which trains TS diffusion model by leveraging the self-imputing capability of the diffusion model and globally aggregating from clients' knowledge through data distillation and clients' synthetic data. The central component of FedTDD is a distiller at the coordinator that first is pre-trained on the public datasets and then periodically fine-tuned by the aggregated intermediate synthetic data from the clients. Clients keep their personalized TS diffusion models and train them with local data and synthetic data of the latest distiller periodically. Our extensive evaluation across five datasets shows that FedTDD effectively overcomes the hurdle of feature partition and temporal misalignment, achieving improvements of up to 79.4% and 62.8% over local training on Context-FID and Correlational scores, while delivering performance comparable to centralized baselines.

### 6 REPRODUCIBILITY AND ETHICS STATEMENT

**Reproducibility**  To ensure the reproducibility of our research, we have open-sourced the code for the various federated learning techniques and the time series diffusion models, as shown in https://anonymous.4open.science/r/FedTDD/. This code is available in a publicly

Table 3: Ablation study for a relatively small number of common features. **Bold** indicates best performance.

| Metric | Method | Stocks | ETTh | MuJoCo | Energy | fMRI |
|--------|--------|--------|------|--------|--------|------|
| Context-FID | Centralized* | 0.682 +/- 0.106 | 0.281 +/- 0.040 | 0.782 +/- 0.138 | 0.533 +/- 0.082 | 1.737 +/- 0.125 |
| | Centralized | 3.733 +/- 0.959 | 11.54 +/- 3.894 | 14.68 +/- 4.263 | 13.17 +/- 3.035 | 15.34 +/- 4.789 |
| | Local | 1.982 +/- 0.234 | 0.824 +/- 0.105 | 0.660 +/- 0.100 | 0.844 +/- 0.127 | 1.220 +/- 0.098 |
| | Pre-trained | 0.738 +/- 0.142 | 0.316 +/- 0.032 | 0.547 +/- 0.099 | 0.381 +/- 0.066 | **1.178 +/- 0.104** |
| | FedTDD | **0.680 +/- 0.123** | **0.267 +/- 0.036** | **0.510 +/- 0.072** | **0.331 +/- 0.051** | 1.196 +/- 0.098 |
| Correlational | Centralized* | 0.061 +/- 0.043 | 0.253 +/- 0.094 | 1.989 +/- 0.247 | 5.231 +/- 1.294 | 7.900 +/- 0.384 |
| | Centralized | 0.697 +/- 0.168 | 0.523 +/- 0.095 | 2.317 +/- 0.597 | 5.781 +/- 0.924 | 31.35 +/- 4.923 |
| | Local | 0.091 +/- 0.052 | 0.167 +/- 0.057 | 1.079 +/- 0.196 | 1.984 +/- 0.594 | **4.929 +/- 0.395** |
| | Pre-trained | 0.028 +/- 0.027 | **0.132 +/- 0.054** | 1.115 +/- 0.233 | 1.795 +/- 0.577 | 5.033 +/- 0.323 |
| | FedTDD | **0.025 +/- 0.022** | 0.137 +/- 0.064 | **1.060 +/- 0.209** | **1.737 +/- 0.282** | 5.005 +/- 0.317 |
| Discriminative | Centralized* | 0.136 +/- 0.091 | 0.199 +/- 0.061 | 0.297 +/- 0.108 | 0.230 +/- 0.080 | 0.422 +/- 0.074 |
| | Centralized | 0.475 +/- 0.041 | 0.469 +/- 0.020 | 0.479 +/- 0.026 | 0.494 +/- 0.010 | 0.484 +/- 0.023 |
| | Local | 0.300 +/- 0.116 | 0.208 +/- 0.070 | 0.190 +/- 0.088 | 0.241 +/- 0.071 | **0.398 +/- 0.058** |
| | Pre-trained | 0.119 +/- 0.088 | 0.116 +/- 0.067 | 0.163 +/- 0.088 | 0.130 +/- 0.058 | 0.418 +/- 0.050 |
| | FedTDD | **0.112 +/- 0.097** | **0.107 +/- 0.078** | **0.157 +/- 0.104** | **0.120 +/- 0.067** | 0.412 +/- 0.057 |
| Predictive | Centralized* | 0.040 +/- 0.000 | 0.127 +/- 0.003 | 0.112 +/- 0.015 | 0.292 +/- 0.009 | 0.137 +/- 0.004 |
| | Centralized | 0.168 +/- 0.025 | 0.196 +/- 0.027 | 0.198 +/- 0.049 | 0.314 +/- 0.052 | 0.223 +/- 0.029 |
| | Local | 0.084 +/- 0.038 | 0.114 +/- 0.009 | 0.069 +/- 0.010 | 0.199 +/- 0.007 | **0.130 +/- 0.005** |
| | Pre-trained | 0.028 +/- 0.007 | 0.108 +/- 0.004 | 0.063 +/- 0.007 | 0.190 +/- 0.005 | 0.132 +/- 0.005 |
| | FedTDD | **0.028 +/- 0.005** | **0.107 +/- 0.005** | **0.062 +/- 0.006** | 0.186 +/- 0.004 | **0.130 +/- 0.005** |

accessible repository under an anonymous account. Furthermore, all experiments conducted as part of this study utilized publicly available datasets.

**Ethics statement** In this research on federated learning over temporally and feature-misaligned datasets, we carefully considered both the positive and potential negative effects. By leveraging federated learning, we enhance data privacy, as no raw data is centralized, reducing risks of sensitive information exposure. However, misaligned datasets may introduce biases that impact model fairness, which we actively worked to mitigate. While our work aims to advance privacy-preserving AI, we acknowledge potential trade-offs in performance and fairness, which are transparently reported to inform future research and applications.

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

# A    NOTATIONS

Table 4 shows the notations used in FedTDD.

Table 4: Descriptions of notations used in FedTDD.

| Variable | Description |
|---|---|
| **Data Variables** | |
| $\mathbf{X}^i, \mathbf{X}^i_{j,k}$ | Time series dataset of client $i$ and data point at time index $j$, feature $k$ |
| $\tilde{\mathbf{X}}^{\text{pub}}_{j,k}, \tilde{\mathbf{X}}^i_{\text{train}_{j,k}}$ | The denoised estimate from $\mathcal{D}$ and $\mathcal{U}$ |
| $\tilde{\mathbf{X}}^i$ | Synthetic dataset generated by client $i$ |
| **Mask Variables** | |
| $\mathbf{M}^i$ | Binary mask matrix indicating observed data for client $i$ |
| **Model Variables** | |
| $\mathcal{D}$ | Global distiller model trained by the coordinator |
| $\mathcal{U}^i$ | Local imputer model of client $i$ |
| FFT | Fast Fourier Transform function |
| **Indices and Parameters** | |
| $N$ | Number of clients participating in federated learning |
| $T^i$ | Number of time steps in client $i$'s dataset |
| $C^i$ | Number of channels (features) in client $i$'s dataset |
| $r$ | Round index in iterative training (from 1 to $R$) |
| $R$ | Total number of training rounds |
| $\alpha$ | Hyperparameter controlling the rate of data expansion |
| $t$ | Diffusion time step |
| $\epsilon$ | Standard Gaussian noise term |
| **Feature Sets** | |
| $\mathcal{F}_{\text{comm}}$ | Set of common features shared across all clients |
| $\mathcal{F}^i$ | Feature set of client $i$ |
| $\mathcal{F}^i_{\text{ex}}$ | Exclusive features specific to client $i$ |
| **Loss Functions** | |
| $\mathcal{L}_{\text{distiller}(\mathcal{D})}$ | Loss function for the distiller model $\mathcal{D}$ |
| $\mathcal{L}_{\text{imputer}(\mathcal{U}^i)}$ | Loss function for imputer model $\mathcal{U}^i$ |

# B    BACKGROUND

In this section, we provide a comprehensive overview of Denoising Diffusion Probabilistic Models (DDPMs) (Ho et al., 2020) and introduce **Diffusion-TS** (Yuan & Qiao, 2024), an extension of DDPMs specifically designed for time series data. We aim to highlight how Diffusion-TS builds upon the foundational principles of DDPMs, particularly focusing on the unique challenges of modeling temporal information inherent in time series data.

## B.1    DENOISING DIFFUSION PROBABILISTIC MODELS

Denoising Diffusion Probabilistic Models (DDPMs) are a class of generative models that have demonstrated remarkable success in modeling complex data distributions, especially in the domain of image generation. The core idea behind DDPMs is to generate new data samples by reversing a predefined noising process. This involves two main stages: a forward diffusion process that progressively adds noise to the data, and a reverse diffusion process that learns to remove the noise to recover the original data distribution.

### B.1.1    FORWARD DIFFUSION PROCESS

The forward diffusion process incrementally corrupts the original data $\mathbf{s}_0 \in \mathbb{R}^d$ through a Markov chain $\mathbf{s}_0, \mathbf{s}_1, \ldots, \mathbf{s}_T$. At each time step $t$, Gaussian noise is added to the data as follows:

$$q(\mathbf{s}_t \mid \mathbf{s}_{t-1}) = \mathcal{N}(\mathbf{s}_t; \sqrt{1 - \beta_t}\, \mathbf{s}_{t-1}, \beta_t\, \mathbf{I}), \tag{7}$$

where $\beta_t \in (0, 1)$ is a predefined variance schedule that controls the amount of noise added at each step, and $\mathbf{I}$ is the identity matrix. The sequence $\{\beta_t\}$ is typically designed so that $\beta_t$ increases over time, ensuring that the data becomes more corrupted as $t$ increases.

The overall forward process can be expressed as:

$$q(\mathbf{s}_{1:T} \mid \mathbf{s}_0) = \prod_{t=1}^{T} q(\mathbf{s}_t \mid \mathbf{s}_{t-1}). \tag{8}$$

As $t$ approaches $T$, the data $\mathbf{s}_t$ becomes increasingly noisy, and in the limit $\mathbf{s}_T$ approaches an isotropic Gaussian distribution $\mathcal{N}(0, \mathbf{I})$. This property is crucial because it allows the reverse process to start from a known simple distribution.

An important feature of the forward process is that we can directly sample $\mathbf{s}_t$ at any time step $t$ from $\mathbf{s}_0$ without simulating all previous steps. By defining $\gamma_t = 1 - \beta_t$ and $\bar{\gamma}_t = \prod_{v=1}^{t} \gamma_v$, we can derive:

$$q(\mathbf{s}_t \mid \mathbf{s}_0) = \mathcal{N}(\mathbf{s}_t; \sqrt{\bar{\gamma}_t}\, \mathbf{s}_0, (1 - \bar{\gamma}_t)\, \mathbf{I}). \tag{9}$$

This expression shows that $\mathbf{s}_t$ is a linear combination of the original data $\mathbf{s}_0$ and Gaussian noise, scaled by the terms $\sqrt{\bar{\gamma}_t}$ and $\sqrt{1 - \bar{\gamma}_t}$, respectively.

Using the reparameterization trick (Kingma, 2013), which is widely used in variational autoencoders, we can write:

$$\mathbf{s}_t = \sqrt{\bar{\gamma}_t}\, \mathbf{s}_0 + \sqrt{1 - \bar{\gamma}_t}\, \boldsymbol{\epsilon} \tag{10}$$

where $\boldsymbol{\epsilon} \sim \mathcal{N}(0, \mathbf{I})$ is a standard Gaussian noise term. This formulation allows us to efficiently compute $\mathbf{s}_t$ and backpropagate gradients during training.

### B.1.2 REVERSE DIFFUSION PROCESS

The goal of the reverse diffusion process is to recover the original data $\mathbf{s}_0$ from the noisy data $\mathbf{s}_T$. This involves learning a reverse Markov chain parameterized by $p_\theta(\mathbf{s}_{t-1} \mid \mathbf{s}_t)$, where $\theta$ represents the model parameters:

$$p_\theta(\mathbf{s}_{t-1} \mid \mathbf{s}_t) = \mathcal{N}(\mathbf{s}_{t-1}; \boldsymbol{\mu}_\theta(\mathbf{s}_t, t), \Sigma_\theta(\mathbf{s}_t, t)). \tag{11}$$

Starting from $\mathbf{s}_T \sim \mathcal{N}(0, \mathbf{I})$, we iteratively sample $\mathbf{s}_{t-1}$ from $p_\theta(\mathbf{s}_{t-1} \mid \mathbf{s}_t)$ until we reach $\mathbf{s}_0$. The functions $\boldsymbol{\mu}_\theta$ and $\Sigma_\theta$ are typically modeled using deep neural networks.

### B.1.3 TRAINING OBJECTIVE AND VARIATIONAL LOWER BOUND

Directly maximizing the data likelihood $\mathbb{E}_{\mathbf{s}_0}[\log p_\theta(\mathbf{s}_0)]$ is intractable due to the high-dimensional integrals involved. Instead, we optimize a variational lower bound (VLB) on the negative log-likelihood:

$$\mathcal{J}_{\text{vlb}} = -\log p_\theta(\mathbf{s}_0 \mid \mathbf{s}_1) + \sum_{t=2}^{T} D_{\text{KL}}\left( q(\mathbf{s}_{t-1} \mid \mathbf{s}_t, \mathbf{s}_0) \,\Big\|\, p_\theta(\mathbf{s}_{t-1} \mid \mathbf{s}_t) \right) + D_{\text{KL}}\left( q(\mathbf{s}_T \mid \mathbf{s}_0) \,\Big\|\, p(\mathbf{s}_T) \right). \tag{12}$$

In this expression, $D_{\text{KL}}$ denotes the Kullback-Leibler divergence between two probability distributions. The first term measures the discrepancy between the true posterior and the model's approximation at the final step, the middle term sums over the discrepancies at each intermediate step, and the last term ensures that the model's prior at $t = T$ matches the known distribution $p(\mathbf{s}_T)$.

### B.1.4 SIMPLIFIED TRAINING OBJECTIVE

Ho et al. (2020) proposed a simplified training objective that focuses on predicting the noise $\boldsymbol{\epsilon}$ added to $\mathbf{s}_0$ at each time step. By reparameterizing the reverse process, they showed that the variational lower bound can be simplified to the following loss function:

$$\mathcal{J}_{\text{simple}} = \mathbb{E}_{t, \mathbf{s}_0, \boldsymbol{\epsilon}} \left[ \|\boldsymbol{\epsilon} - \boldsymbol{\epsilon}_\theta(\mathbf{s}_t, t)\|^2 \right], \tag{13}$$

where $\boldsymbol{\epsilon}_\theta(\mathbf{s}_t, t)$ is the model's estimate of the noise at time step $t$. This loss function is computationally efficient and has been empirically shown to produce high-quality generative models.

### B.1.5 RELATION TO SCORE MATCHING

Estimating $\epsilon$ is closely related to estimating the score function, which is the gradient of the log probability density function with respect to the data. Specifically, the score function $\nabla_{\mathbf{s}_t} \log q(\mathbf{s}_t \mid \mathbf{s}_0)$ can be expressed as:

$$\nabla_{\mathbf{s}_t} \log q(\mathbf{s}_t \mid \mathbf{s}_0) = -\frac{1}{1 - \bar{\gamma}_t} \left( \mathbf{s}_t - \sqrt{\bar{\gamma}_t}\, \mathbf{s}_0 \right) = -\frac{1}{\sqrt{1 - \bar{\gamma}_t}}\, \epsilon. \tag{14}$$

This shows that predicting $\epsilon$ is equivalent to learning the scaled score function of the noisy data distribution.

### B.1.6 SAMPLING PROCEDURE

After training, new data samples can be generated by starting from $\mathbf{s}_T \sim \mathcal{N}(0, \mathbf{I})$ and iteratively applying the learned reverse transitions. The sampling process at each step $t$ is given by:

$$\mathbf{s}_{t-1} = \frac{1}{\sqrt{\gamma_t}} \left( \mathbf{s}_t - \frac{1 - \gamma_t}{\sqrt{1 - \bar{\gamma}_t}}\, \epsilon_\theta(\mathbf{s}_t, t) \right) + \sigma_t\, \mathbf{z}, \tag{15}$$

where $\sigma_t$ is a hyperparameter controlling the randomness in the sampling process, and $\mathbf{z} \sim \mathcal{N}(0, \mathbf{I})$. This equation updates $\mathbf{s}_t$ towards the estimated mean while adding some noise to maintain stochasticity.

## B.2 DIFFUSION-TS: INTERPRETABLE DIFFUSION FOR TIME SERIES

While DDPMs have been successful in modeling high-dimensional data such as images, they do not explicitly account for the unique characteristics of time series data, which often include temporal dependencies, trends, and seasonal patterns. **Diffusion-TS** (Yuan & Qiao, 2024) extends the DDPM framework to address these challenges by incorporating an interpretable decomposition architecture specifically designed for time series analysis.

### B.2.1 ADAPTED DIFFUSION FRAMEWORK FOR TIME SERIES

In Diffusion-TS, both the forward and reverse diffusion processes are adapted to capture the temporal structures inherent in time series data.

**Forward process** The forward process remains similar to that of standard DDPMs but is tailored to handle sequential data. The time series data $\mathbf{s}_0 \in \mathbb{R}^d$, where $d$ represents the sequence length, is gradually corrupted using a variance schedule $\delta_t$:

$$q(\mathbf{s}_t \mid \mathbf{s}_{t-1}) = \mathcal{N}\left( \mathbf{s}_t; \sqrt{1 - \delta_t}\, \mathbf{s}_{t-1}, \delta_t\, \mathbf{I} \right), \tag{16}$$

where $\delta_t \in (0, 1)$ controls the noise level at each diffusion step. The cumulative product $\bar{\gamma}_t = \prod_{v=1}^{t} \gamma_v$, with $\gamma_t = 1 - \delta_t$, allows for direct computation of $\mathbf{s}_t$ from $\mathbf{s}_0$.

**Reverse process** In the reverse process, Diffusion-TS modifies the parameterization by directly predicting an estimate of the original time series $\hat{\mathbf{s}}_0(\mathbf{s}_t, t; \theta)$. The reverse transition is formulated as:

$$\mathbf{s}_{t-1} = \frac{\sqrt{\bar{\gamma}_{t-1}}\delta_t}{1 - \bar{\gamma}_t}\, \hat{\mathbf{s}}_0(\mathbf{s}_t, t; \theta) + \frac{\sqrt{\gamma_t}(1 - \bar{\gamma}_{t-1})}{1 - \bar{\gamma}_t}\, \mathbf{s}_t + \sigma_t\, \mathbf{z}_t, \tag{17}$$

where $\sigma_t = \sqrt{\delta_t}$ and $\mathbf{z}_t \sim \mathcal{N}(0, \mathbf{I})$. By predicting $\hat{\mathbf{s}}_0$ directly, the model focuses on reconstructing the original time series, which is crucial for capturing temporal dependencies.

### B.2.2 DECOMPOSITION-BASED MODEL ARCHITECTURE

To effectively model time series data, Diffusion-TS employs a decomposition-based architecture that explicitly models trend and seasonality components, inspired by classical time series decomposition methods.

**Trend synthesis**  The trend component $\mathbf{v}_{\text{tr}}$ captures the long-term progression in the data. It is synthesized using polynomial regression (Oreshkin et al., 2019; Desai et al., 2021):

$$\mathbf{v}_{\text{tr}} = \sum_{i=1}^{D} \left( \mathbf{C} \cdot \text{Linear}\left( \mathbf{w}_{\text{tr}}^{(i)} \right) + \mathbf{x}_{\text{tr}}^{(i)} \right), \tag{18}$$

where $D$ is the number of decoder layers in the transformer architecture, $\mathbf{C} = [\mathbf{1}, \mathbf{c}, \mathbf{c}^2, \dots, \mathbf{c}^p]$ is a matrix of polynomial basis vectors, with $\mathbf{c} = [0, 1, \dots, d-1]^T/d$ representing normalized time indices and $p$ being the polynomial degree, $\mathbf{w}_{\text{tr}}^{(i)}$ are learnable weights for the $i$-th layer's trend component, and $\mathbf{x}_{\text{tr}}^{(i)}$ is the mean output from the $i$-th decoder block, capturing the average behavior. This approach allows the model to capture smooth and continuous changes over time.

**Seasonality and residual synthesis**  The seasonality component $\mathbf{s}_{\text{seas}}^{(i)}$ captures repeating patterns or cycles in the data using Fourier series (De Livera et al., 2011; Woo et al., 2022). The amplitude $A_i^{(k)}$ and phase $\Phi_i^{(k)}$ of the $k$-th frequency component are computed as follows:

$$A_i^{(k)} = \left| \mathcal{F}\left( \mathbf{w}_{\text{seas}}^{(i)} \right)_k \right|, \quad \Phi_i^{(k)} = \arg\left( \mathcal{F}\left( \mathbf{w}_{\text{seas}}^{(i)} \right)_k \right), \tag{19}$$

$$\mathbf{s}_{\text{seas}}^{(i)} = \sum_{k=1}^{K} A_i^{(k)} \cos\left( 2\pi f_k\, \mathbf{c} + \Phi_i^{(k)} \right), \tag{20}$$

where $\mathcal{F}$ denotes the Fourier Transform applied to the weights $\mathbf{w}_{\text{seas}}^{(i)}$, $A_i^{(k)}$ and $\Phi_i^{(k)}$ are the amplitude and phase of the $k$-th frequency component, $K$ is the number of significant frequencies selected based on their amplitudes, and $f_k$ is the $k$-th frequency component. The residual component $\mathbf{r}$ accounts for any remaining patterns or noise not captured by the trend and seasonality components. The final reconstructed time series is obtained by combining these components:

$$\hat{\mathbf{s}}_0(\mathbf{s}_t, t; \theta) = \mathbf{v}_{\text{tr}} + \sum_{i=1}^{D} \mathbf{s}_{\text{seas}}^{(i)} + \mathbf{r}. \tag{21}$$

### B.2.3  Fourier-Based Training Objective

To ensure the model effectively learns and disentangles the time series components, Diffusion-TS introduces a loss function that operates in both the time and frequency domains:

$$\mathcal{J}_\theta = \mathbb{E}_{t, \mathbf{s}_0} \left[ w_t \left( \lambda_1 \| \mathbf{s}_0 - \hat{\mathbf{s}}_0(\mathbf{s}_t, t; \theta) \|^2 + \lambda_2 \| \text{FFT}(\mathbf{s}_0) - \text{FFT}(\hat{\mathbf{s}}_0(\mathbf{s}_t, t; \theta)) \|^2 \right) \right], \tag{22}$$

where $w_t = \frac{\lambda \gamma_t (1 - \bar{\gamma}_t)}{\delta_t^2}$ is a weighting term that emphasizes learning at larger diffusion steps, with $\lambda$ being a small constant. $\lambda_1$ and $\lambda_2$ are hyperparameters that balance the contributions of the time-domain loss and the frequency-domain loss, and FFT denotes the Fast Fourier Transform, which transforms the time series into the frequency domain. By incorporating the frequency-domain loss, the model is encouraged to accurately capture periodic components, improving its ability to model seasonality. Using mean squared error loss to train this denoising model is refer to (Ho et al., 2020)

### B.2.4  Handling Temporal Information Differently

Diffusion-TS differs from standard DDPMs in several key aspects tailored for time series data. First, the model explicitly separates the time series into trend, seasonality, and residual components, enhancing interpretability and modeling capabilities. Second, by incorporating a loss in the frequency domain, the model emphasizes the learning of periodic patterns, which are prevalent in time series data. Lastly, the use of an encoder-decoder transformer architecture allows the model to capture long-range temporal dependencies and complex sequential patterns. These adaptations enable Diffusion-TS to effectively model the intricate temporal dynamics present in time series data, addressing the limitations of standard DDPMs that do not explicitly account for temporal structures.

Table 5: Results on multiple time series datasets when **PR** is 0.25. **Bold** indicates best performance.

| Metric | Method | Stocks | ETTh | MuJoCo | Energy | fMRI |
|---|---|---|---|---|---|---|
| Context-FID | Centralized* | 0.389 +/- 0.045 | 0.179 +/- 0.022 | 0.591 +/- 0.129 | 0.500 +/- 0.084 | 1.239 +/- 0.095 |
| | Centralized | 4.255 +/- 0.808 | 7.405 +/- 2.354 | 8.273 +/- 2.918 | 12.75 +/- 4.210 | 11.37 +/- 2.654 |
| | Local | 1.372 +/- 0.253 | 1.240 +/- 0.126 | 0.956 +/- 0.213 | 1.838 +/- 0.265 | 1.491 +/- 0.096 |
| | Pre-trained | 0.483 +/- 0.066 | 0.300 +/- 0.039 | 0.567 +/- 0.147 | 0.488 +/- 0.077 | **1.267 +/- 0.086** |
| | FedTDD | **0.367 +/- 0.048** | **0.230 +/- 0.033** | **0.562 +/- 0.136** | **0.468 +/- 0.071** | 1.322 +/- 0.087 |
| Correlational | Centralized* | 0.056 +/- 0.052 | 0.203 +/- 0.061 | 1.535 +/- 0.245 | 4.159 +/- 0.933 | 6.603 +/- 0.277 |
| | Centralized | 0.472 +/- 0.283 | 0.255 +/- 0.050 | 1.968 +/- 0.221 | 4.951 +/- 0.487 | 16.73 +/- 1.074 |
| | Local | 0.120 +/- 0.069 | 0.215 +/- 0.078 | 1.103 +/- 0.181 | 2.999 +/- 0.763 | **5.031 +/- 0.238** |
| | Pre-trained | 0.069 +/- 0.052 | **0.134 +/- 0.046** | 1.084 +/- 0.173 | 2.258 +/- 0.553 | 5.196 +/- 0.203 |
| | FedTDD | **0.044 +/- 0.038** | 0.140 +/- 0.058 | **1.058 +/- 0.148** | **2.159 +/- 0.521** | 5.297 +/- 0.261 |
| Discriminative | Centralized* | 0.124 +/- 0.100 | 0.103 +/- 0.052 | 0.320 +/- 0.064 | 0.239 +/- 0.055 | 0.383 +/- 0.054 |
| | Centralized | 0.473 +/- 0.032 | 0.430 +/- 0.020 | 0.437 +/- 0.036 | 0.489 +/- 0.009 | 0.433 +/- 0.041 |
| | Local | 0.331 +/- 0.107 | 0.285 +/- 0.050 | **0.257 +/- 0.096** | 0.351 +/- 0.048 | **0.391 +/- 0.044** |
| | Pre-trained | 0.178 +/- 0.107 | 0.121 +/- 0.065 | 0.261 +/- 0.084 | **0.137 +/- 0.051** | 0.419 +/- 0.041 |
| | FedTDD | **0.095 +/- 0.098** | **0.106 +/- 0.049** | 0.258 +/- 0.056 | 0.153 +/- 0.047 | 0.409 +/- 0.044 |
| Predictive | Centralized* | 0.034 +/- 0.001 | 0.126 +/- 0.008 | 0.090 +/- 0.010 | 0.283 +/- 0.007 | 0.128 +/- 0.005 |
| | Centralized | 0.042 +/- 0.003 | 0.210 +/- 0.016 | 0.187 +/- 0.045 | 0.278 +/- 0.024 | 0.183 +/- 0.023 |
| | Local | 0.037 +/- 0.001 | 0.116 +/- 0.007 | 0.042 +/- 0.005 | 0.201 +/- 0.008 | 0.128 +/- 0.004 |
| | Pre-trained | **0.036 +/- 0.000** | 0.102 +/- 0.004 | 0.038 +/- 0.003 | **0.171 +/- 0.003** | 0.129 +/- 0.004 |
| | FedTDD | 0.036 +/- 0.001 | **0.100 +/- 0.005** | **0.036 +/- 0.004** | **0.171 +/- 0.003** | **0.128 +/- 0.003** |

### B.2.5 Conditional Generation for Time Series Applications

For practical applications like imputation (filling missing values) and forecasting (predicting future values), Diffusion-TS extends its framework to conditional generation. Given observed data $\mathbf{y}$, the model aims to generate samples consistent with this data. To achieve this, Diffusion-TS employs gradient-based guidance during the sampling process. The estimated time series $\hat{\mathbf{s}}_0$ is adjusted at each diffusion step using:

$$\tilde{\mathbf{s}}_0(\mathbf{s}_t, t; \theta) = \hat{\mathbf{s}}_0(\mathbf{s}_t, t; \theta) + \eta \nabla_{\mathbf{s}_t} \left( \|\mathbf{y} - \hat{\mathbf{s}}_0(\mathbf{s}_t, t; \theta)\|^2 + \gamma \log p(\mathbf{s}_{t-1} \mid \mathbf{s}_t) \right), \quad (23)$$

where $\eta$ is a hyperparameter controlling the strength of the gradient guidance, $\gamma$ balances the trade-off between fitting the observed data and adhering to the learned data distribution, and $\log p(\mathbf{s}_{t-1} \mid \mathbf{s}_t)$ represents the model's prior, ensuring that the generated data remains realistic. By iteratively refining $\tilde{\mathbf{s}}_0$ using gradient information, the model generates samples that not only match the observed data but also maintain the overall temporal coherence and patterns learned during training.

## C Additional Experimental Results

### C.1 Limited Public Data Availability

We analyze the performance of FedTDD against other baselines when the public data is limited by reducing the public ratio (**PR**) from 0.5 to 0.25. As shown in Table 5, FedTDD maintains its performance (ETTh and MuJoCo declined slightly) even with limited public data, and the results are notably better compared to those with a **PR** of 0.5 in Table 2. This improvement is due to the increased amount of client data as **PR** decreases, which allows for better time series generation as clients train on more data. Nevertheless, FedTDD continues to struggle with the fMRI dataset, as discussed in Section 4.1.

### C.2 Abundance of Sequences with Incomplete Data

We assess the robustness of FedTDD in handling a large number of sequences with missing values across individual clients by decreasing the split ratio (**SR**) from 0.5 to 0.25. As explained in Section 4, decreasing the **SR** increases the missingness across both common and exclusive features in

Table 6: Results on multiple time series datasets when **SR** is 0.25. **Bold** indicates best performance.

| Metric | Method | Stocks | ETTh | MuJoCo | Energy | fMRI |
|---|---|---|---|---|---|---|
| Context-FID | Centralized* | 0.682 +/- 0.106 | 0.281 +/- 0.040 | 0.782 +/- 0.138 | 0.533 +/- 0.082 | 1.737 +/- 0.125 |
| | Centralized | 4.373 +/- 1.392 | 8.080 +/- 2.113 | 10.35 +/- 3.222 | 8.951 +/- 2.331 | 15.01 +/- 5.081 |
| | Local | 3.010 +/- 0.438 | 1.237 +/- 0.144 | 0.851 +/- 0.110 | 1.202 +/- 0.156 | 2.055 +/- 0.187 |
| | Pre-trained | 0.802 +/- 0.122 | 0.413 +/- 0.069 | 0.642 +/- 0.078 | 0.472 +/- 0.066 | **1.372 +/- 0.106** |
| | FedTDD | **0.699 +/- 0.104** | **0.365 +/- 0.058** | **0.636 +/- 0.076** | **0.433 +/- 0.058** | 1.462 +/- 0.118 |
| Correlational | Centralized* | 0.061 +/- 0.043 | 0.253 +/- 0.094 | 1.989 +/- 0.247 | 5.231 +/- 1.294 | 7.900 +/- 0.384 |
| | Centralized | 0.375 +/- 0.227 | 0.329 +/- 0.076 | 2.135 +/- 0.454 | 5.686 +/- 0.552 | 17.63 +/- 2.684 |
| | Local | 0.192 +/- 0.143 | 0.241 +/- 0.076 | 1.311 +/- 0.199 | 3.576 +/- 0.678 | 6.439 +/- 0.529 |
| | Pre-trained | 0.082 +/- 0.063 | 0.157 +/- 0.052 | 1.365 +/- 0.254 | 2.958 +/- 0.762 | **6.112 +/- 0.329** |
| | FedTDD | **0.069 +/- 0.064** | **0.153 +/- 0.069** | **1.308 +/- 0.240** | **2.909 +/- 0.905** | 6.211 +/- 0.372 |
| Discriminative | Centralized* | 0.136 +/- 0.091 | 0.199 +/- 0.061 | 0.297 +/- 0.108 | 0.230 +/- 0.080 | 0.422 +/- 0.074 |
| | Centralized | 0.459 +/- 0.051 | 0.471 +/- 0.025 | 0.457 +/- 0.042 | 0.498 +/- 0.003 | 0.486 +/- 0.019 |
| | Local | 0.290 +/- 0.122 | 0.285 +/- 0.079 | 0.227 +/- 0.117 | 0.341 +/- 0.077 | **0.420 +/- 0.062** |
| | Pre-trained | 0.200 +/- 0.130 | 0.174 +/- 0.079 | 0.232 +/- 0.098 | 0.150 +/- 0.068 | 0.440 +/- 0.050 |
| | FedTDD | **0.176 +/- 0.127** | **0.153 +/- 0.071** | **0.188 +/- 0.085** | **0.149 +/- 0.065** | 0.431 +/- 0.054 |
| Predictive | Centralized* | 0.040 +/- 0.000 | 0.127 +/- 0.003 | 0.112 +/- 0.015 | 0.292 +/- 0.009 | 0.137 +/- 0.004 |
| | Centralized | 0.044 +/- 0.006 | 0.237 +/- 0.002 | 0.218 +/- 0.044 | 0.464 +/- 0.033 | 0.374 +/- 0.020 |
| | Local | **0.042 +/- 0.003** | 0.121 +/- 0.013 | **0.049 +/- 0.007** | 0.202 +/- 0.011 | 0.131 +/- 0.006 |
| | Pre-trained | 0.044 +/- 0.002 | 0.108 +/- 0.003 | 0.058 +/- 0.006 | **0.177 +/- 0.006** | 0.131 +/- 0.004 |
| | FedTDD | 0.044 +/- 0.003 | **0.105 +/- 0.004** | 0.050 +/- 0.005 | 0.181 +/- 0.006 | **0.130 +/- 0.003** |

client data. In Table 6, we observe a slight decline in some metrics, but overall FedTDD performs well and even outperforms Local and Pre-trained baselines in terms of Context-FID, Correlational and Discriminative scores across most datasets. Again, FedTDD continues to underperform on the fMRI dataset, as mentioned in Section 4.1.

## C.3 IMBALANCED DATA DISTRIBUTIONS

We now evaluate the performance of FedTDD against other baselines presented in Table 7 using imbalanced partitioned datasets, where each partition may contain distinct data distributions. FedTDD generally performs better in terms of Context-FID score across most datasets except for MuJoCo and fMRI. This highlights the strength of FedTDD in maintaining synthetic data distribution with real data in most imbalanced dataset scenarios.

**Performance of Local training** The results show that training locally without communication with the coordinator is comparable to all other methods, particularly outperforming Centralized* and Centralized in most metrics. Notably, it achieves the best Discriminative score across datasets, especially in MuJoCo, Energy and fMRI, indicating that locally trained models generate more realistic synthetic data. In contrast, FedTDD and other baselines (excluding Local) perform below expectations, especially FedTDD and Pre-trained. This is primarily due to the evaluation of synthetic data using test data with different distributions corresponding to each client, while other approaches can generate more generalized synthetic data. For instance, the clients in FedTDD and Pre-trained learn different data distributions during the coordinator's imputation process. FedTDD performs even worse than Pre-trained due to the additional data aggregation and fine-tuning process. Moreover, Centralized* and Centralized approaches train all the data including public data at once, which leads to the worst performance. As a result, local models might be more suited for generating synthetic data that is distributed closer to the respective test data.

## C.4 AGGREGATION STRATEGY

We evaluate the effectiveness of FedTDD on the Stocks and fMRI datasets by comparing our proposed aggregation strategy with three other aggregation settings: (1) 1 : 0, e.g. fine-tune with 100 public samples and 0 synthetic samples, (2) 1 : 1, e.g. fine-tune with 100 public samples and 100

Table 7: Results on multiple imbalanced partitioned datasets. **Bold** indicates best performance.

| Metric | Method | Stocks | ETTh | MuJoCo | Energy | fMRI |
|---|---|---|---|---|---|---|
| Context-FID | Centralized* | 1.911 +/- 0.198 | 1.736 +/- 0.226 | 1.933 +/- 0.376 | 3.361 +/- 0.510 | 2.311 +/- 0.178 |
| | Centralized | 5.057 +/- 1.228 | 11.45 +/- 2.225 | 11.39 +/- 4.001 | 11.78 +/- 3.678 | 15.01 +/- 3.275 |
| | Local | 1.355 +/- 0.180 | 0.813 +/- 0.097 | 0.703 +/- 0.117 | 0.961 +/- 0.079 | 1.676 +/- 0.144 |
| | Pre-trained | 0.568 +/- 0.062 | 0.320 +/- 0.046 | **0.644 +/- 0.089** | 0.583 +/- 0.066 | **1.540 +/- 0.108** |
| | FedTDD | **0.463 +/- 0.066** | **0.258 +/- 0.035** | 0.675 +/- 0.088 | **0.570 +/- 0.057** | 1.549 +/- 0.103 |
| Correlational | Centralized* | 0.153 +/- 0.105 | 0.383 +/- 0.093 | 2.234 +/- 0.316 | 9.474 +/- 1.755 | 7.985 +/- 0.449 |
| | Centralized | 0.900 +/- 0.270 | 0.441 +/- 0.079 | 2.403 +/- 0.533 | 6.711 +/- 0.766 | 19.19 +/- 2.149 |
| | Local | 0.162 +/- 0.117 | 0.173 +/- 0.070 | 1.282 +/- 0.204 | 3.095 +/- 0.874 | **6.075 +/- 0.351** |
| | Pre-trained | **0.090 +/- 0.083** | **0.131 +/- 0.056** | **1.260 +/- 0.257** | 2.515 +/- 0.699 | 6.171 +/- 0.461 |
| | FedTDD | 0.096 +/- 0.095 | 0.151 +/- 0.055 | 1.274 +/- 0.216 | **2.464 +/- 0.733** | 6.137 +/- 0.351 |
| Discriminative | Centralized* | 0.322 +/- 0.109 | 0.371 +/- 0.058 | 0.364 +/- 0.070 | 0.444 +/- 0.024 | 0.446 +/- 0.036 |
| | Centralized | 0.472 +/- 0.036 | 0.487 +/- 0.015 | 0.467 +/- 0.032 | 0.492 +/- 0.011 | 0.476 +/- 0.024 |
| | Local | 0.255 +/- 0.110 | 0.242 +/- 0.066 | **0.150 +/- 0.098** | **0.265 +/- 0.078** | **0.277 +/- 0.081** |
| | Pre-trained | **0.150 +/- 0.121** | 0.186 +/- 0.072 | 0.320 +/- 0.120 | 0.415 +/- 0.049 | 0.451 +/- 0.045 |
| | FedTDD | 0.169 +/- 0.096 | **0.178 +/- 0.090** | 0.324 +/- 0.097 | 0.416 +/- 0.042 | 0.443 +/- 0.047 |
| Predictive | Centralized* | 0.085 +/- 0.002 | 0.226 +/- 0.028 | 0.118 +/- 0.021 | 0.302 +/- 0.017 | 0.141 +/- 0.007 |
| | Centralized | 0.089 +/- 0.010 | 0.327 +/- 0.006 | 0.346 +/- 0.042 | 0.303 +/- 0.033 | 0.247 +/- 0.050 |
| | Local | 0.079 +/- 0.008 | 0.128 +/- 0.016 | **0.057 +/- 0.012** | 0.180 +/- 0.006 | 0.134 +/- 0.005 |
| | Pre-trained | **0.069 +/- 0.003** | 0.108 +/- 0.007 | 0.065 +/- 0.006 | **0.171 +/- 0.004** | 0.133 +/- 0.005 |
| | FedTDD | 0.070 +/- 0.003 | **0.107 +/- 0.007** | 0.067 +/- 0.006 | 0.171 +/- 0.005 | **0.132 +/- 0.004** |

Table 8: Results of four aggregation strategies on Stocks and fMRI datasets. **Bold** indicates best performance.

| Metric | Setting | Stocks | fMRI |
|---|---|---|---|
| Context-FID | $1 : 0$ | 1.012 +/- 0.163 | **1.419 +/- 0.077** |
| | $1 : 1$ | 1.090 +/- 0.242 | 2.078 +/- 0.184 |
| | $1 : [0, 1]$ | **0.675 +/- 0.087** | 1.785 +/- 0.128 |
| | $1 : \alpha[0, 1]$ (ours) | **0.675 +/- 0.087** | 1.459 +/- 0.099 |
| Correlational | $1 : 0$ | 0.084 +/- 0.093 | 6.061 +/- 0.277 |
| | $1 : 1$ | 0.078 +/- 0.071 | 6.318 +/- 0.309 |
| | $1 : [0, 1]$ | **0.058 +/- 0.050** | 6.241 +/- 0.408 |
| | $1 : \alpha[0, 1]$ (ours) | **0.058 +/- 0.050** | **6.017 +/- 0.364** |
| Discriminative | $1 : 0$ | **0.148 +/- 0.124** | 0.426 +/- 0.054 |
| | $1 : 1$ | 0.172 +/- 0.093 | **0.412 +/- 0.061** |
| | $1 : [0, 1]$ | 0.185 +/- 0.105 | 0.414 +/- 0.062 |
| | $1 : \alpha[0, 1]$ (ours) | 0.185 +/- 0.105 | 0.414 +/- 0.051 |
| Predictive | $1 : 0$ | **0.041 +/- 0.001** | 0.135 +/- 0.004 |
| | $1 : 1$ | 0.042 +/- 0.002 | 0.133 +/- 0.004 |
| | $1 : [0, 1]$ | **0.041 +/- 0.001** | **0.131 +/- 0.005** |
| | $1 : \alpha[0, 1]$ (ours) | **0.041 +/- 0.001** | 0.133 +/- 0.004 |

synthetic samples, **(3)** $1 : [0, 1]$, e.g. fine-tune with 100 public samples and increase the synthetic samples linearly from 0 to 100, **(4)** $1 : \alpha[0, 1]$ **(ours)**, e.g. fine-tune with 100 public samples and increase the synthetic samples linearly from 0 to 100 by a factor $\alpha$. We explore $\alpha$ values of $[0.1, 0.5, 1.0]$, where $\alpha = 1.0$ is equivalent to setting (3). As shown in Table 8, our proposed strategy (4) achieves the best results across most metrics on Stocks dataset, particularly with $\alpha = 1.0$. Although our approach does not perform as well on the fMRI dataset, it effectively prevents further deterioration of results. We list the $\alpha$ used for each dataset in Appendix D.5.

## D  EXPERIMENT DETAILS

### D.1  HARDWARE AND SOFTWARE

We run experiments on an Intel(R) Core(TM) i7-14700KF processor and an NVIDIA GeForce RTX 4090 GPU with CUDA version 12.5. The operating system for the setup is Ubuntu 22.04.4 LTS. Flower framework and PyTorch are used to simulate the distributed experiments by creating multiple models corresponding to individual clients.

### D.2  DATASETS

Table 9 shows the statistics of all benchmark datasets. We provide the number of rows and features.

Table 9: Statistics of datasets.

| Dataset | #Rows | #Features | Source |
|---------|-------|-----------|--------|
| Stocks | 3773 | 6 | https://finance.yahoo.com/quote/GOOG |
| ETTh | 17420 | 7 | https://github.com/zhouhaoyi/ETDataset |
| MuJoCo | 10000 | 14 | https://github.com/deepmind/dm_control |
| Energy | 19711 | 28 | https://archive.ics.uci.edu/ml/datasets |
| fMRI | 10000 | 50 | https://www.fmrib.ox.ac.uk/datasets |

### D.3  EVALUATION METRICS

**Context-FID score**  Jeha et al. (2022) introduced the Context-FID score, which is an adaptation of existing Fréchet inception distance (FID) used for evaluating the similarity between real and synthetic time series distributions. Instead of the Inception model used for the image feature extractor, Context-FID leverages a time series embedding model called TS2Vec (Yue et al., 2022). The authors demonstrated that models with lower Context-FID generally perform better in downstream tasks, such as achieving a strong correlation between Context-FID and the forecasting performance of generative model. In conclusion, a lower Context-FID score indicates greater similarity between the real and synthetic distributions.

**Correlational score**  Liao et al. (2020) estimates the covariance of the $i^{th}$ and $j^{th}$ feature of time series using the following formula:

$$\text{cov}_{i,j} = \frac{1}{\mathcal{T}} \sum_{t=1}^{\mathcal{T}} Y_i^t Y_i^t - \left( \frac{1}{\mathcal{T}} \sum_{t=1}^{\mathcal{T}} Y_i^t \right) \left( \frac{1}{\mathcal{T}} \sum_{t=1}^{\mathcal{T}} Y_j^t \right). \tag{24}$$

To quantify the correlation between real and synthetic data, we compute the following metric:

$$\frac{1}{10} \sum_{i,j}^{d} \left| \frac{\text{cov}_{i,j}^{real}}{\sqrt{\text{cov}_{i,i}^{real} \text{cov}_{j,j}^{real}}} - \frac{\text{cov}_{i,j}^{synth}}{\sqrt{\text{cov}_{i,i}^{synth} \text{cov}_{j,j}^{synth}}} \right|, \tag{25}$$

**Discriminative score**  The discriminative score is determined by the formula $|\text{accuracy} - 0.5|$, which measures the ability of the model to differentiate between real and synthetic data. A lower score indicates better performance as the model struggles to distinguish between the two, implying higher similarity. For consistency, we adapt the experimental setup of TimeGAN (Yoon et al., 2019) with a 2-layer GRU-based neural network as the classifier.

**Predictive score**  The predictive score is computed as the MAE between the predicted and actual values on the test data. Again, we use the experimental configuration of TimeGAN (Yoon et al., 2019) with a 2-layer GRU-based neural network for sequence prediction.

## D.4 MISSING SCENARIO

Figure 5 shows the missing scenario we used in all experiments.

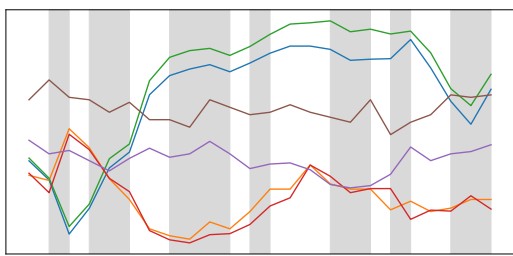

Figure 5: We consider random missing strategy on multivariate time series. The white background represents the conditional ground truth, while the grey background represents the time steps of the particular channel that must be imputed.

## D.5 HYPERPARAMETERS

In Table 10, we list the hyperparameter settings for all datasets. To evaluate the effectiveness of FedTDD, 80% of each client's data is used for training, with the remaining 20% reserved for testing.

Table 10: Hyperparameters

| Parameter | Stocks | ETTh | MuJoCo | Energy | fMRI |
|---|---|---|---|---|---|
| Attention heads | 4 | 4 | 4 | 4 | 4 |
| Attention head dimension | 16 | 16 | 16 | 24 | 24 |
| Encoder layers | 2 | 3 | 3 | 4 | 4 |
| Decoder layers | 2 | 2 | 2 | 3 | 4 |
| Batch size | 64 | 128 | 128 | 64 | 64 |
| Alpha, $\alpha$ | 1.0 | 0.1 | 0.1 | 0.5 | 0.1 |
| Timesteps / Sampling steps | 500 | 500 | 1000 | 1000 | 1000 |
| Pre-trained training steps | 10000 | 18000 | 14000 | 25000 | 25000 |

## E VISUALIZATIONS

Figure 6 to 8 present the performance of time series synthesis in three different visualization techniques. **Principal component analysis (PCA)**, **t-distributed Stochastic Neighbor Embedding (t-SNE)** and **kernel density estimation (KDE)** are used to visualize how well the generated synthetic data distributions align the real data distributions (PCA and t-SNE project the data in 2-dimensional space). The figures show that FedTDD achieves significantly better performance with greater overlap and closer similarity between the real and synthetic samples.

## F SYNTHETIC SAMPLES

Figure 9 to 13 demonstrate synthetic time series generated unconditionally by FedTDD and Local approach against real time series data. The generated synthetic samples from FedTDD closely resemble the real samples across most datasets. In each figure, the first row corresponds to the first client, the second row to the second client, and so forth. A maximum of 4 features are selected randomly for each client.

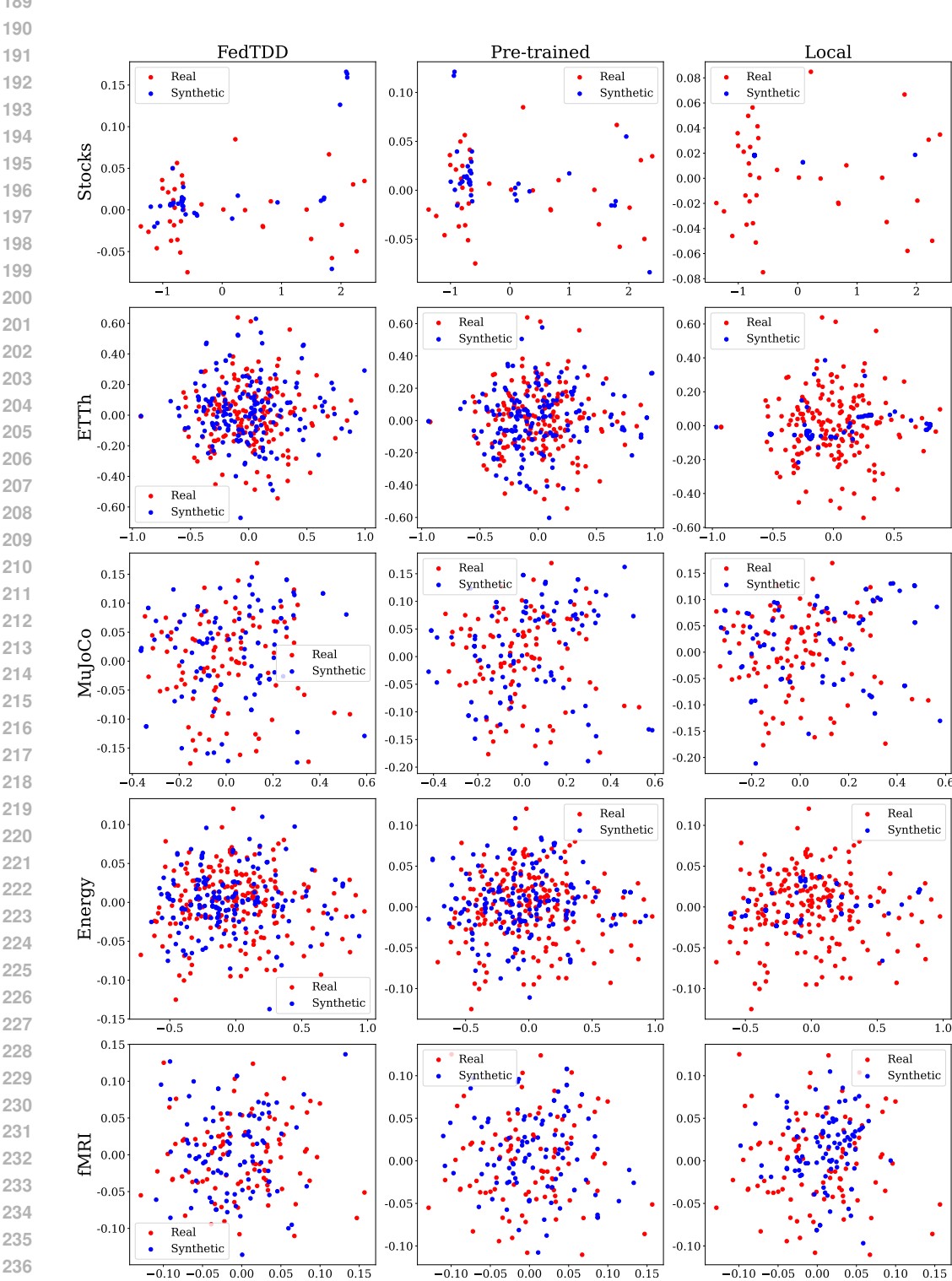

Figure 6: PCA plots of real and synthetic time series generated by one representative client from FedTDD, Pre-trained and Local on all datasets.

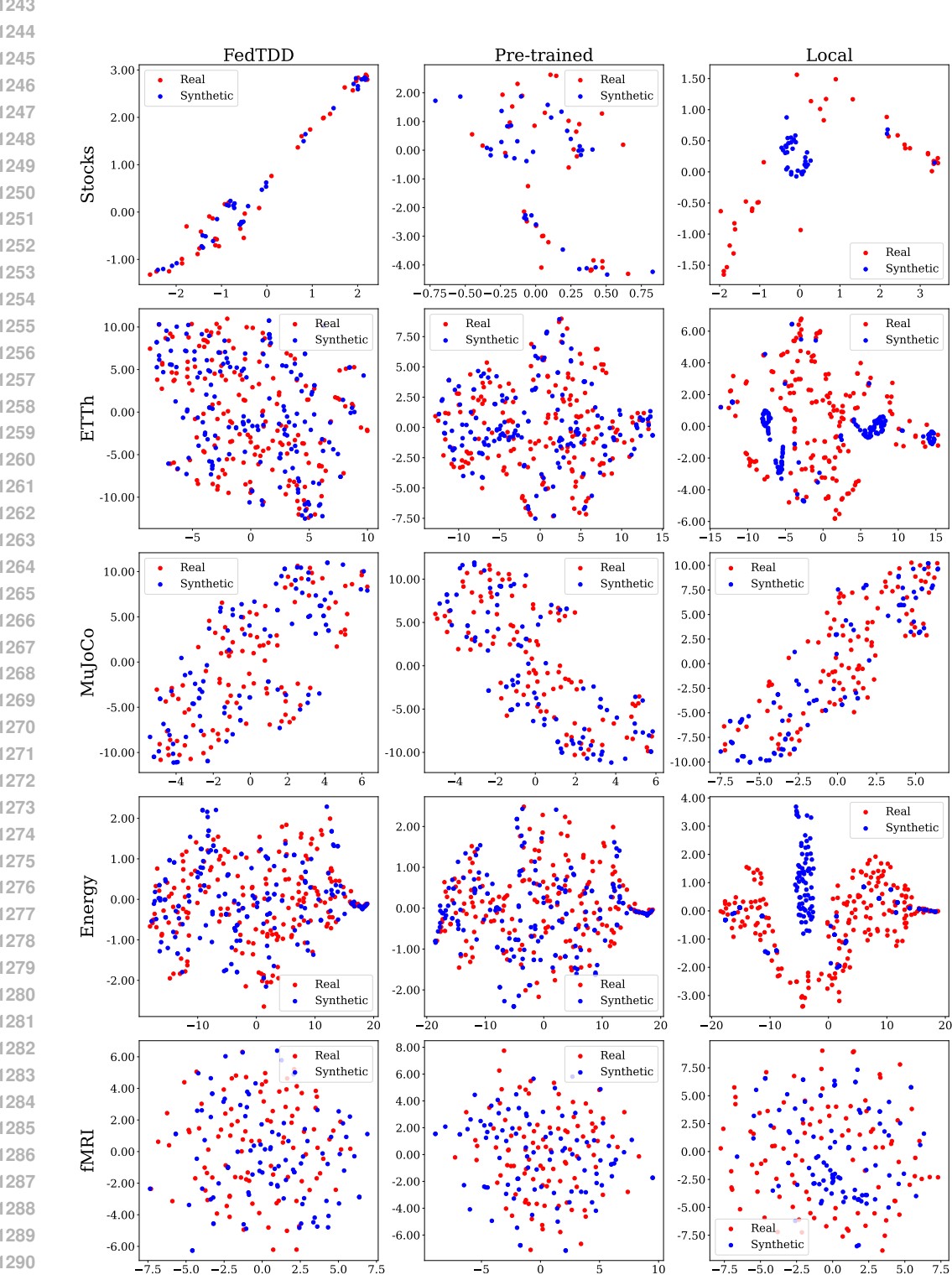

Figure 7: t-SNE plots of real and synthetic time series generated by one representative client from FedTDD, Pre-trained and Local on all datasets.

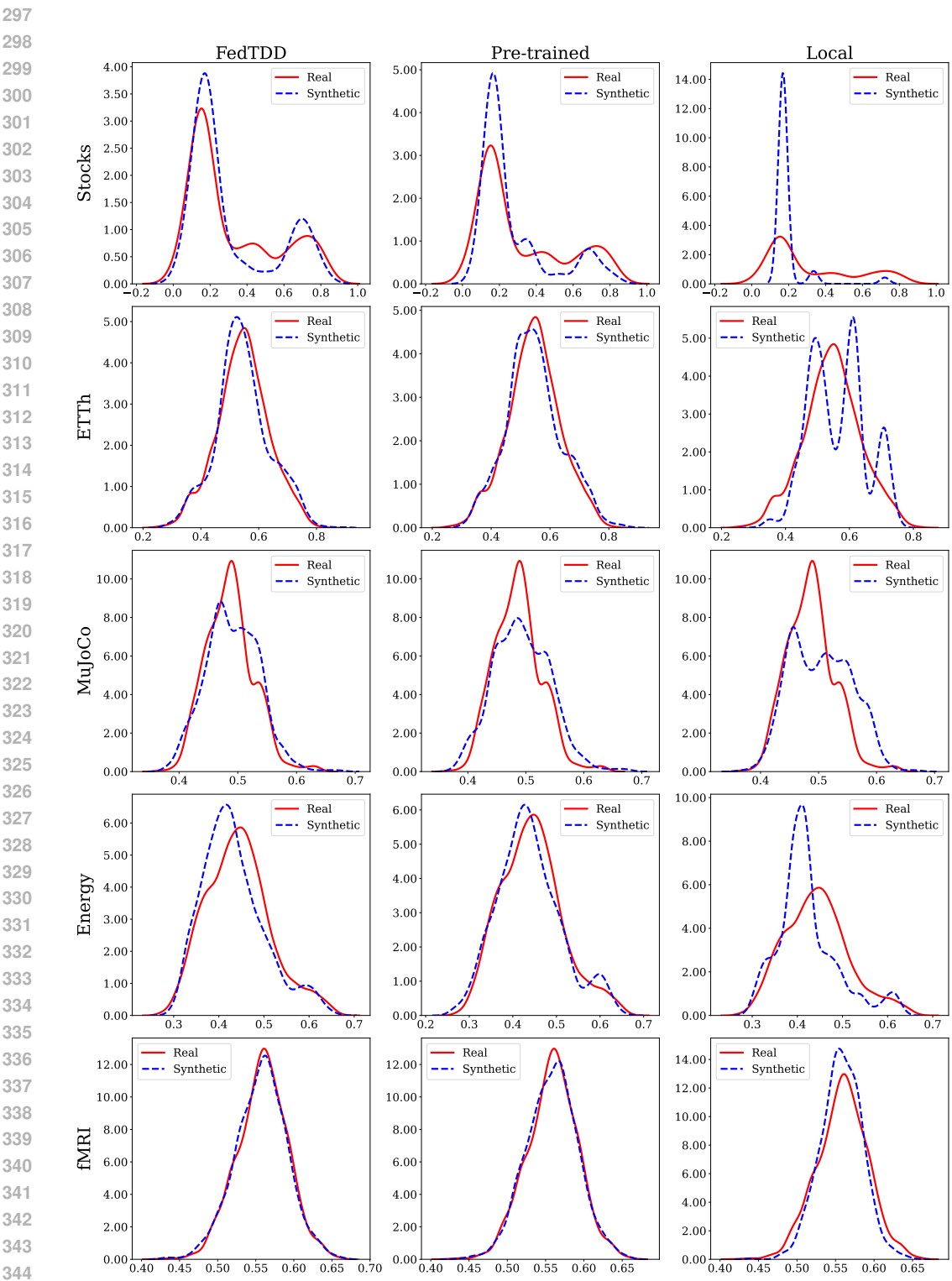

Figure 8: KDE plots of real and synthetic time series generated by one representative client from FedTDD, Pre-trained and Local on all datasets. The y-axis of the plots represents the data density estimation.

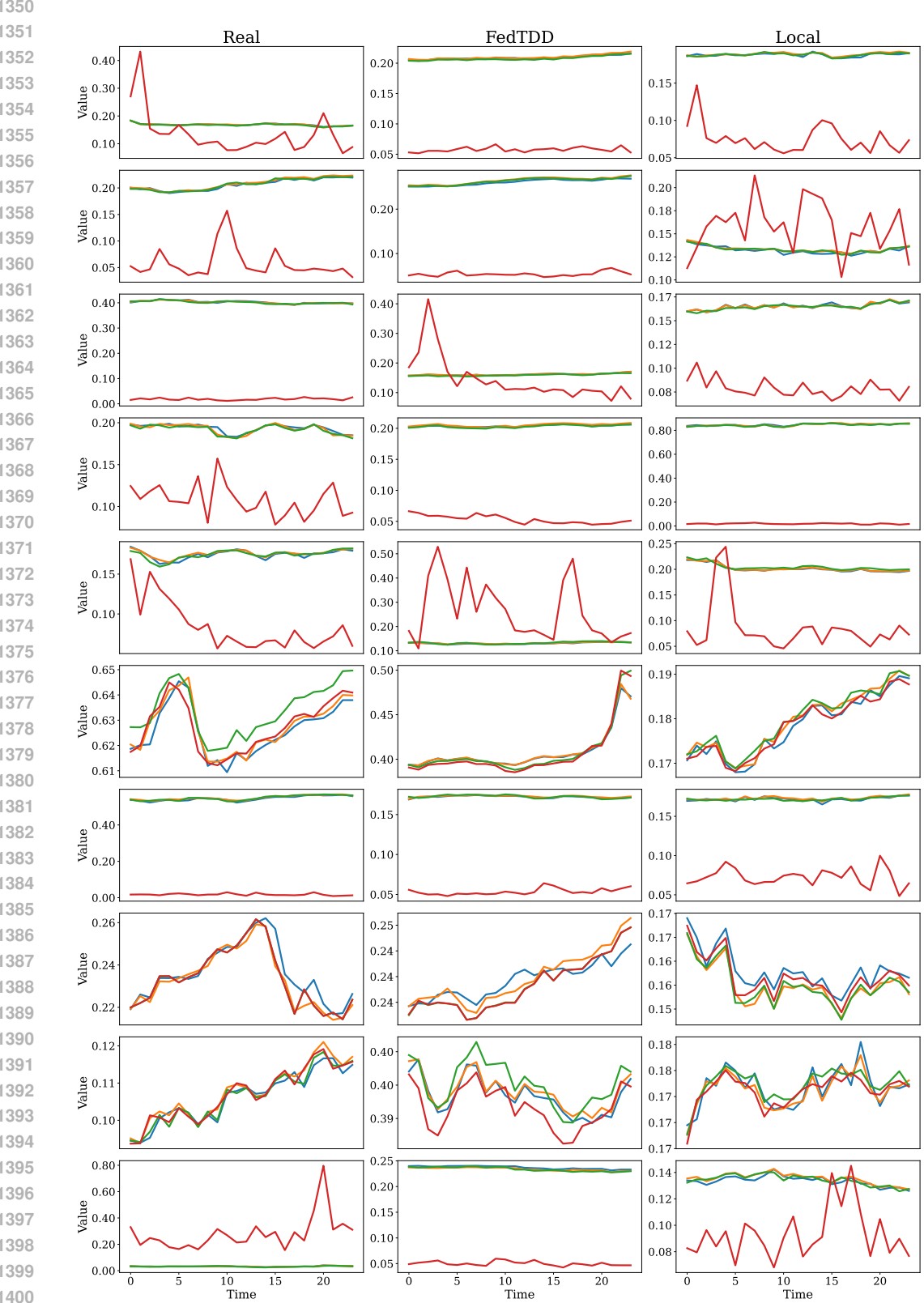

Figure 9: Real samples and synthetic samples generated by FedTDD and Local for the Stocks dataset.

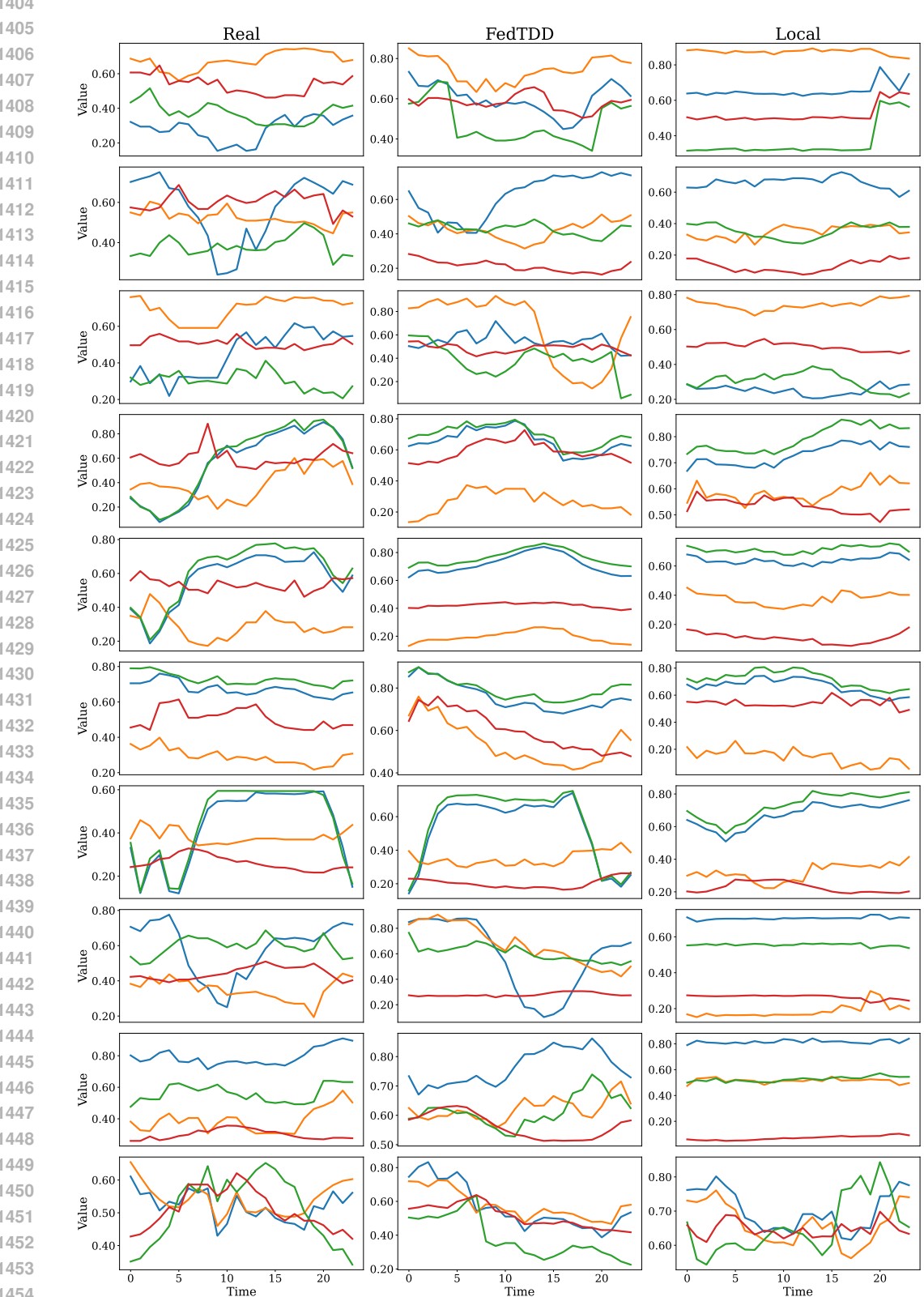

Figure 10: Real samples and synthetic samples generated by FedTDD and Local for the ETTh dataset.

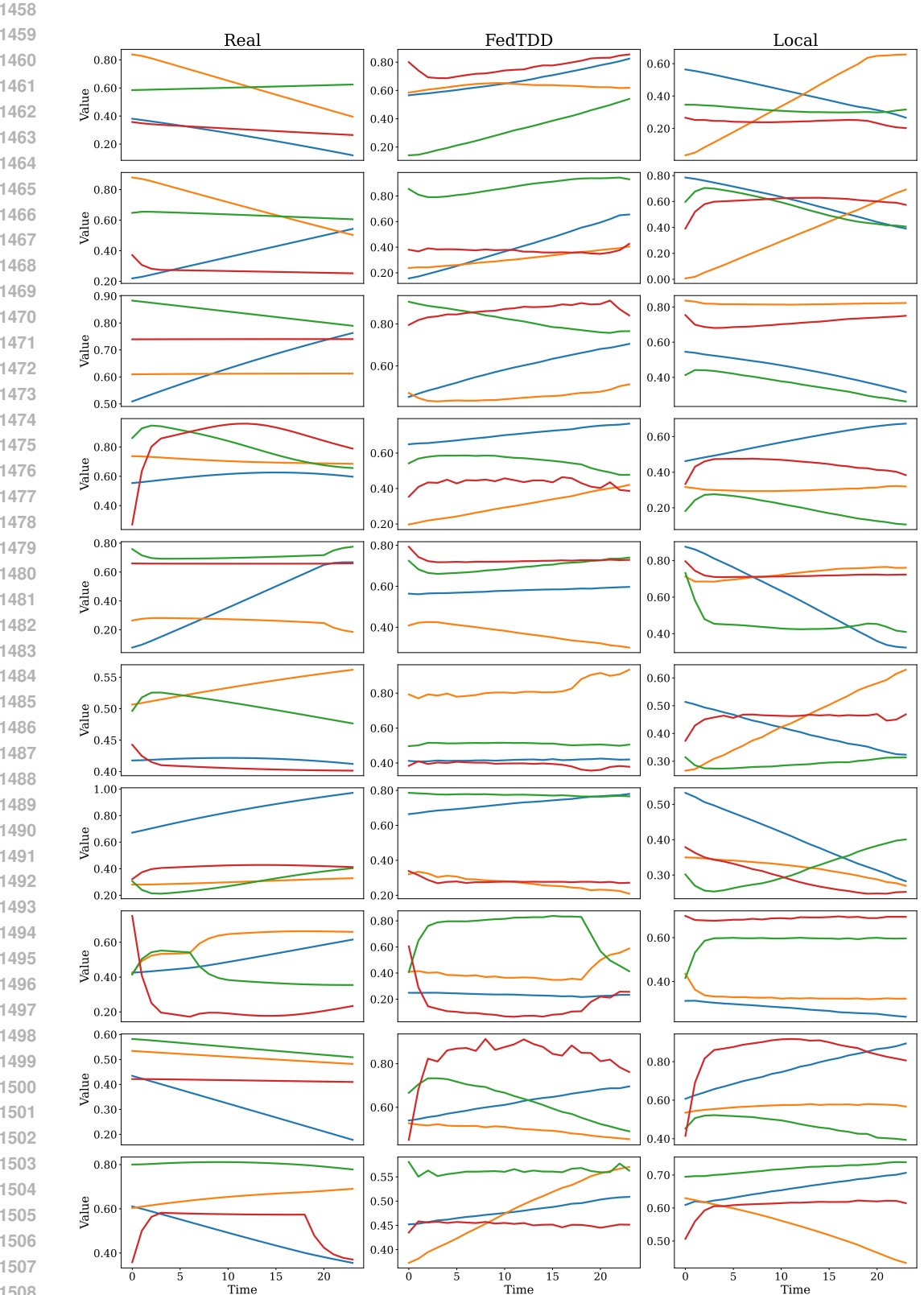

Figure 11: Real samples and synthetic samples generated by FedTDD and Local for the MuJoCo dataset.

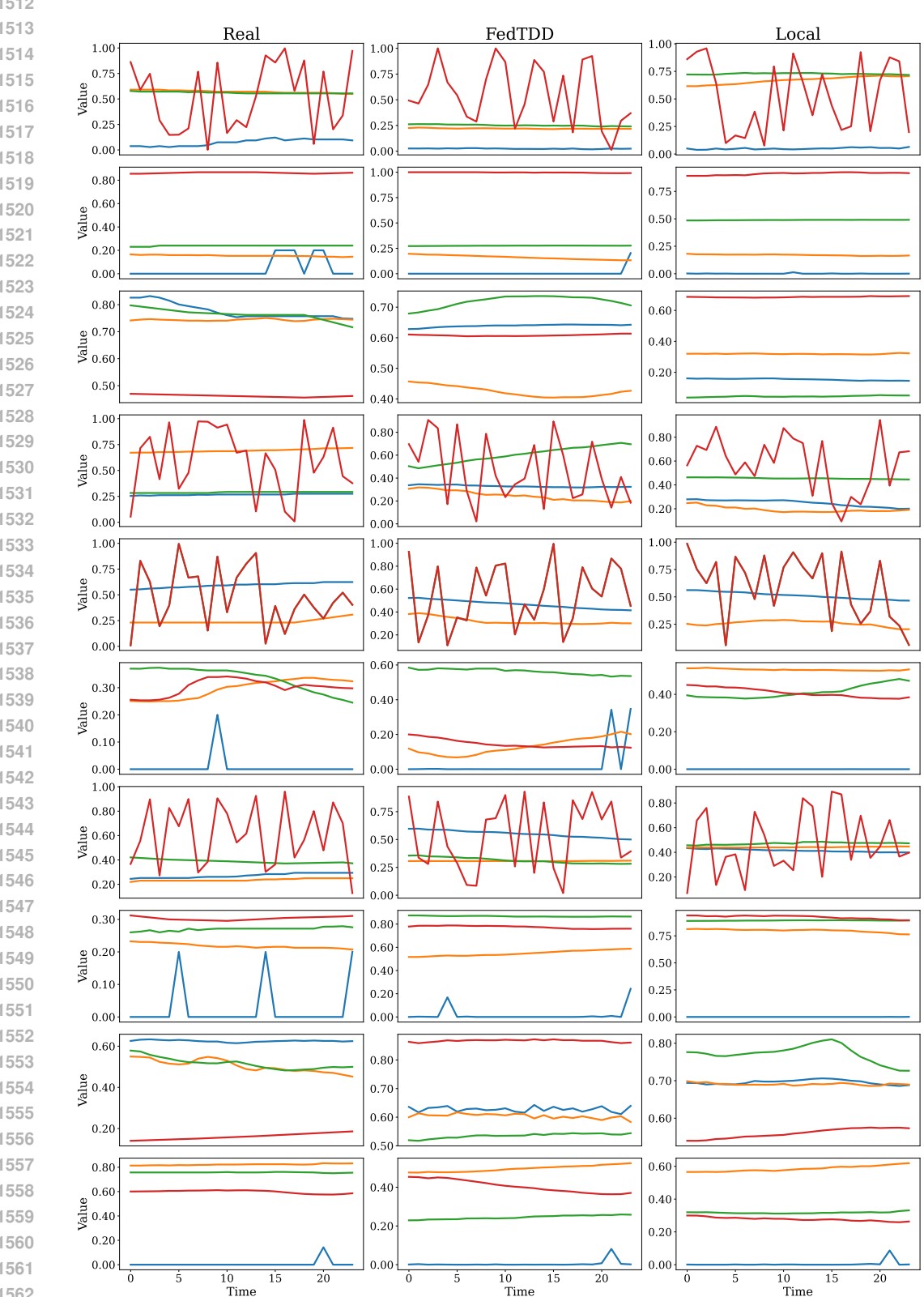

Figure 12: Real samples and synthetic samples generated by FedTDD and Local for the Energy dataset.

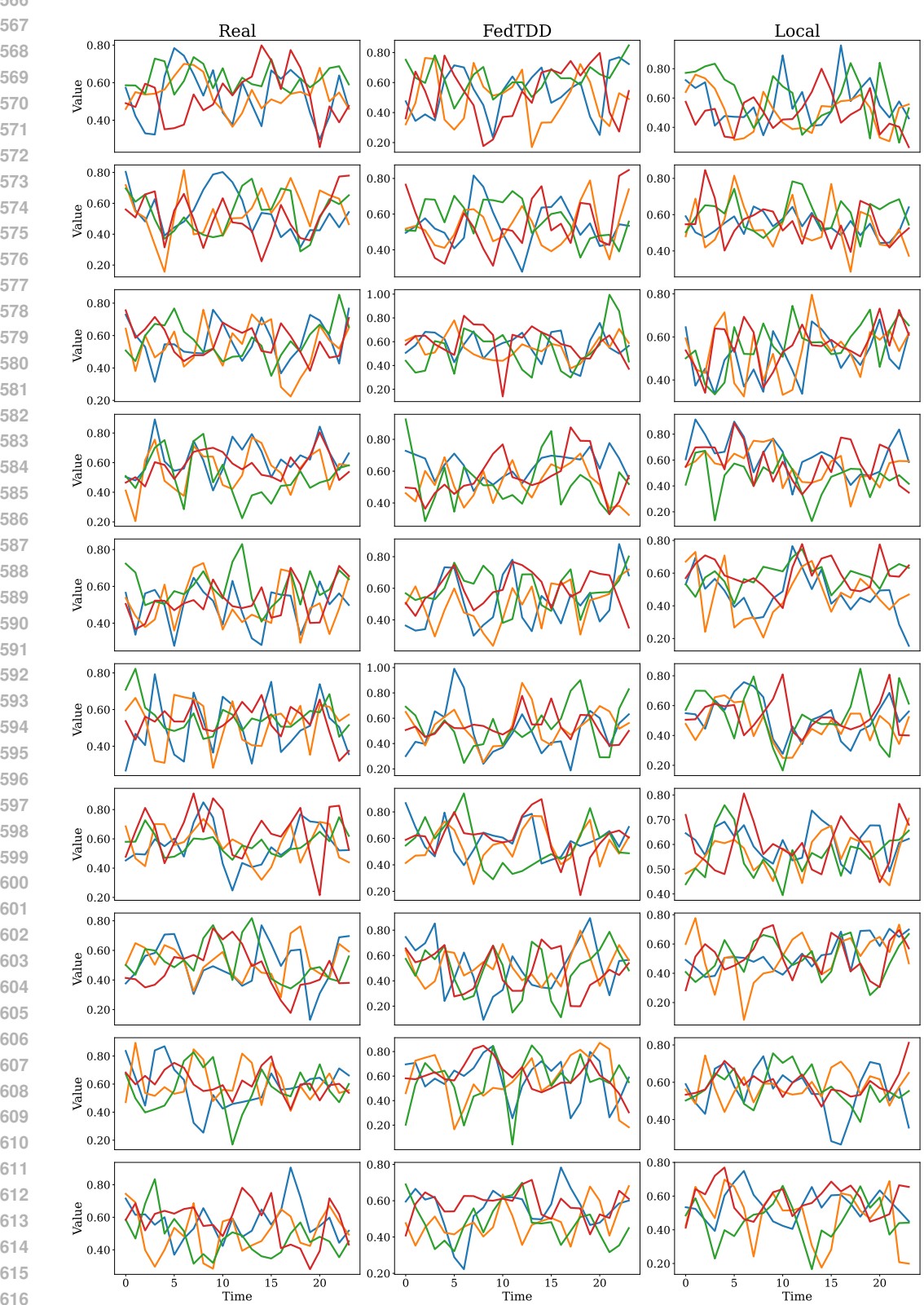

Figure 13: Real samples and synthetic samples generated by FedTDD and Local for the fMRI dataset.

