# OpenReview forum: "Federated Time Series Generation on Feature and Temporally Misaligned Data"
_ICLR.cc/2025/Conference — Submitted to ICLR 2025_

### Official Review · Reviewer_qtP3 · 2024-10-30

**Soundness:** 3
**Presentation:** 3
**Contribution:** 3
**Rating:** 6
**Confidence:** 5

**Summary:**

The paper proposes FedTDD, a novel federated time series generation framework designed to address the challenges of feature and temporal misalignment across clients. This method introduce a GAN-inspired adversarial mechanism between the global distiller and local imputers, enabling collaboration among clients with heterogeneous data by synthesizing data to bridge these discrepancies, preserving clients' privacy.

**Strengths:**

1.	FedTDD offers an innovative solution to key federated learning challenges by addressing both feature and temporal misalignment, issues often overlooked by previous methods. By enabling clients to generate and share synthetic data instead of raw data, it effectively preserves privacy in a novel and promising way.
2.	The well-designed data distillation framework introduces a novel approach that is particularly advantageous for privacy-sensitive fields like healthcare.
3.	The paper is well-written and easy to follow.

**Weaknesses:**

1.	The experimental setup focuses on synthetic data quality metrics (Context-FID, Correlational Score, Discriminative Score, and Predictive Score) instead of clients' average imputation accuracy or downstream task performance. This raises concerns about the actual effectiveness of the proposed method.
2.	The experiments utilize only the diffusion model, without assessing the FedTDD framework’s generalizability across other established generative models like GANs or VAEs.
3.	The study does not evaluate communication efficiency. Since FedTDD involves transferring synthetic datasets rather than model parameters, it may be significantly more resource-intensive, especially in large-scale or bandwidth-constrained settings.
4.	The paper could strengthen its evaluations by including baselines that combine SOTA methods from both vertical and horizontal federated learning.
5.	The released repository lacks a well-prepared README.

**Questions:**

1.	Given that the evaluation metrics focus on the quality of synthetic data, could you clarify why you chose not to evaluate the clients' average imputation accuracy or downstream task performance? How can we be assured that FedTDD effectively enhances client-side utility beyond synthetic data quality?
2.	Have you considered testing FedTDD with other established generative models like GANs or VAEs to demonstrate its generalizability? If not, could you discuss any anticipated challenges or limitations in using these models within the FedTDD framework?
3.	Could you provide more insights into the communication overhead, especially in large-scale or bandwidth-limited settings?

---

### Official Review · Reviewer_xWMU · 2024-11-01

**Soundness:** 2
**Presentation:** 2
**Contribution:** 2
**Rating:** 5
**Confidence:** 5

**Summary:**

The paper studies an important problem of time series generation and proposes a federated learning based method to collaboratively train local time series generation models enabling privacy.

**Strengths:**

1.	The paper is well-organized and in a good logic.
2.	The paper proposes a federated time series diffusion model for decentralized time series generation, which considers temporal misalignment.
3.	Experiments show the effectiveness of the proposed method to some extent.

**Weaknesses:**

1. The focus of federated learning is to protect privacy by keeping data decentralized. The proposed method requires to maintain data with common features in the server (or coordinator), which raises concerns regarding privacy. It would be better to provide a strategy to ensure privacy protection when uploading common features to server with a theoretical guarantee. Even if the data is synthetic from the raw data, existing attack-based inverse methods can easily recover the raw sensitive data.
2. Typically, the inference stage of the diffusion model requires more training time, which hurts the efficiency. However, we often consider edge devices as clients in federated learning, which only have limited computation capabilities. It would be better to design a lightweight module for the clients. In addition, it is suggested to include theoretical time and space complexities analysis of the proposed method. Moreover, it is encouraged to compare the training time, FLOPs, and parameters of the proposed methods and baselines (e.g., TimeGAN, TimeVAE, CSDI).
3. It would be more interesting to assess the effect of different time series generation diffusion models by replacing the distiller, such as TimeGrad, CSDI, SSSD, TSDiff, and Diffusion-TS.
4. It would be promising to transform existing SOTA time series generation methods (e.g., TimeGAN, TimeVAE, S4 [1], Time weaver [2]) into their federated version and compare them with the proposed FedTDD. Please refer to this benchmark [3].
[1]. Deep Latent State Space Models for Time-Series Generation, ICLR 2023.
[2]. Time weaver: A conditional time series generation model, ICML 2024.
[3]. TSGBench: Time Series Generation Benchmark, PVLDB 2024.
5. Data heterogeneity is a big issue in federated learning, especially for time series. It is encouraged to include a specific module to address data heterogeneity across clients.

**Questions:**

Please see the weaknesses.

---

### Official Review · Reviewer_oGju · 2024-11-02

**Soundness:** 4
**Presentation:** 4
**Contribution:** 3
**Rating:** 6
**Confidence:** 4

**Summary:**

This paper studies a time series imputation problem under federated learning setting. To address the temporal and feature misalignment of dataset, the paper proposes FedTDD, a federated learning framework for time series generation from client’s distinct features and public dataset. Different from traditional federated learning, FedTDD learns the correlations among clients’ time series through the exchange of synthetic outputs rather than model parameters between distiller and clients. The comprehensive experiments demonstrate the effectiveness of FedTDD.

**Strengths:**

+ The research problem of distributed time series generation is interesting and practical.
+ FedTDD introduces an innovative federated learning framework by exchanging synthetic data exchange rather than model parameters, leading to enhanced privacy and imputation performance.
+ The experimental results show significant improvements over of FedTDD compared to baselines.

**Weaknesses:**

- The baselines in experiments are relatively straightforward.
- It would be better if the paper shows more experimental results on parameter analysis.

**Questions:**

Please refer to the weaknesses part.

---

### Official Review · Reviewer_3qDg · 2024-11-04

**Soundness:** 3
**Presentation:** 3
**Contribution:** 3
**Rating:** 5
**Confidence:** 3

**Summary:**

The paper addresses the challenge of synthesizing time series data in a federated context, where the time series data at the clients may be misaligned either in terms of time or in terms of features.  The synthesis takes place via learning which does not require sharing of the raw data.

**Strengths:**

Addresses a gap in the literature through its ability to handle both feature misalignment and temporal misalignment, not just one of these.  In this respect the contribution is original.

The paper is generally clearly written and presented.

Consistent improvements generated over baseline methods and the performance of the FedTDD method is close to a centralized approach and better than the local approach.

Method leverages diffusion models in an interesting way to facilitate generation.

**Weaknesses:**

A substantive assessment of the weaknesses of the paper. Focus on constructive and actionable insights on how the work could improve towards its stated goals. Be specific, avoid generic remarks. For example, if you believe the contribution lacks novelty, provide references and an explanation as evidence; if you believe experiments are insufficient, explain why and exactly what is missing, etc

It is hard to assess significance since experimentally evaluating the approach requires many assumptions to be made.     A range of decisions have been made for configurations used in experiments.   Not immediately clear whether these are reasonable and whether they provide adequate insight into performance across the entire configuration space.  E.g. setting the number of common features to 50% or 25%.    Could a real application be referenced for which this would be a realistic setting?


Complexity and scalability of the approach is not analysed.  Diffusion models have a reputation for sometimes being slow to train.   Is this the case for FedTDD?

Lots of metrics are analyzed, but it is hard to get an overall idea of performance

Requires 8(?) hyperparameters to be chosen.

**Questions:**

What is significance of the colors in figure 4?  Is each color a different feature?   What message  do you wish the reader to get from looking at figure 4?

Not clear for me how hyperparameter optimization was performed.  What data was used and how it distinct from other training/test?  Also is the Table 10 list of hyperparameters complete?  Eta and gamma hyperparameters are mentioned in the text but don’t appear in Table 10?

Figure 5 shows the missing configuration.  Can the reasonableness of this be justified?  Why is such a configuration appropriate and realistic?

How does the performance relate to the number of clients.  Can anything be said in general?  Why were 10 clients chosen?

Would you argue it is always beneficial to use FedTDD? Or are there circumstances when the pre-trained approach would be preferable?

Table 1 mentions a number of baselines.  How does FedTDD compare in performance to these when either feature misalignment of temporal misalignment (not both) is present.  Can anything be said?

---

### Comment · Area_Chair_QMn6 · 2024-11-25
**Acknowledge the author responses**

Dear Reviewers,

Thank you very much for your effort. As the discussion period is coming to an end, please acknowledge the author responses and adjust the rating if necessary.

Sincerely,
AC

---

### Meta-Review · Area_Chair_QMn6 · 2024-12-19

**Metareview:**

This paper presents a federated time series diffusion model that jointly learns a synthesizer across clients.  Although the reviewers liked the motivation, presentation, and good performance, they also raised many concerns on the experiment setting and results.  The reviewers were  partially satisfied with the authors' responses during the discussion period.  However, no reviewer strongly supported the acceptance of this paper.  Thus, based on the reviewers' opinions, I recommend a reject.

**Additional Comments On Reviewer Discussion:**

Reviewer xWMU asked for more evaluation using other time-series generation models.  It seems that the authors successfully addressed his/her requests.  Even though I consider this potential increase, the overall rating is sligtly below the acceptance bar, and no reviewer championed the acceptance of this paper.

---

### Decision · Program_Chairs · 2025-01-22

Reject